# Dynamic Mixture of Progressive Parameter-Efficient Expert Library for Lifelong Robot Learning

## Abstract

A generalist agent must continuously learn and adapt throughout its lifetime, achieving efficient forward transfer while minimizing catastrophic forgetting. Previous work within the dominant pretrain-then-finetune paradigm has explored parameter-efficient fine-tuning for single-task adaptation, effectively steering a frozen pretrained model with a small number of parameters. However, in the context of lifelong learning, these methods rely on the impractical assumption of a test-time task identifier and restrict knowledge sharing among isolated adapters. To address these limitations, we propose Dynamic Mixture of Progressive Parameter-Efficient Expert Library (DMPEL) for lifelong robot learning. DMPEL progressively builds a low-rank expert library and employs a lightweight router to dynamically combine experts into an end-to-end policy, enabling flexible and efficient lifelong forward transfer. Furthermore, by leveraging the modular structure of the fine-tuned parameters, we introduce expert coefficient replay, which guides the router to accurately retrieve frozen experts for previously encountered tasks. This technique mitigates forgetting while being significantly more storage- and computation-efficient than experience replay over the entire policy. Extensive experiments on the lifelong robot learning benchmark LIBERO demonstrate that our framework outperforms state-of-the-art lifelong learning methods in success rates during continual adaptation, while utilizing minimal trainable parameters and storage.

## 1 Introduction

A generalist agent should have the capability to learn and adapt continuously throughout its lifetime, usually termed as *lifelong learning* (De Lange et al., 2021; Masana et al., 2022; Mendez & Eaton, 2023; Wang et al., 2024a). The longstanding challenges are enabling *forward transfer* (i.e., leveraging knowledge from previous tasks to quickly adapt to new ones) and avoiding *catastrophic forgetting* (i.e., retaining previously acquired knowledge when learning new tasks) under *limited computational and memory capacity*.

Due to the notorious sample inefficiency of the traditional *tabula rasa* approach in robotics, researchers have recently explored the *pretrain-then-finetune* paradigm (Brohan et al., 2022; 2023; Kim et al., 2025; Black et al., 2024) that originates from the vision and language domains (Bommasani et al., 2021; Oquab et al., 2024; Radford et al., 2021; Brown et al., 2020). Specifically, this paradigm first pretrain a policy, often referred to as a *vision-language-action* (VLA) model, on large-scale datasets, and then fine-tune it for various downstream tasks.

However, in the context of lifelong learning, naively applying full fine-tuning to sequentially arriving robotic tasks can result in severe forgetting and suboptimal performance, failing to meet our requirements. As depicted in Figure 1, prior work in lifelong robot learning can be broadly classified into three distinct approaches. *Replay* methods (Brohan et al., 2023; Lopez-Paz & Ranzato, 2017; Chaudhry et al., 2019; Shin et al., 2017; Xie & Finn, 2022) retain previous data (e.g., vision-language web data, robot pretraining data, and data from seen tasks) and mix it with in-distribution data during training on new tasks. These methods leads to generalizable policies, but also necessitates huge storage space and computational overhead for retaining old samples. *Regularization* methods (Zenke et al., 2017; Kirkpatrick et al., 2017; Li & Hoiem, 2017) balance between new and old tasks

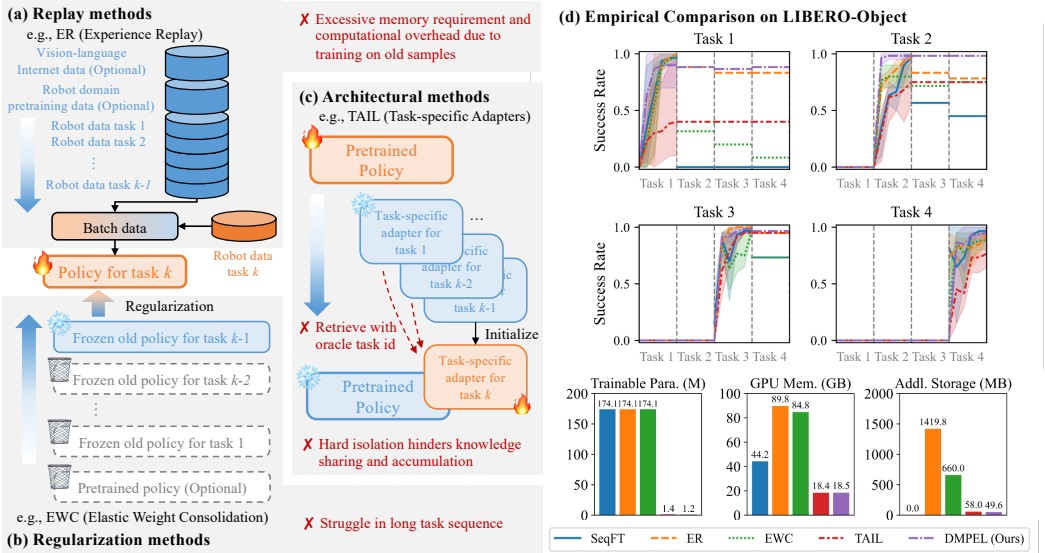

Figure 1: Existing lifelong learning methods: (a) Replay methods; (b) Regularization methods; (c) Architectural methods; (d) Performance on LIBERO-Object. TAIL with LoRA utilizes significantly fewer resources than ER/EWC with FFT and exhibits no forgetting when provided with a task identifier, but demonstrates lower forward transfer. In contrast, DMPEL leverages a low-rank expert library and coefficient replay (CR=5%) to achieve better forward transfer and near-zero forgetting.

by restricting the update of parameters. For example, EWC (Kirkpatrick et al., 2017) require an entire copy of frozen model from the previous task and entail substantial computation in estimating the importance of parameters. In addition, they usually struggle in long task sequence due to plasticity-stability dilemma.

In contrast, *architectural methods* (Rusu et al., 2016; Mallya & Lazebnik, 2018; Mallya et al., 2018; Ge et al., 2024) address catastrophic forgetting by explicitly learning task-specific parameters. The widely adopted *parameter-efficient fine-tuning* (PEFT) techniques (Ding et al., 2023), such as adapters (Houlsby et al., 2019; Chen et al., 2022), low-rank adaptation (LoRA) (Hu et al., 2021), and prompt tuning (Jia et al., 2022; Li & Liang, 2021; Lester et al., 2021), have proven highly effective for steering frozen foundation models by integrating small, learnable modules in single-task adaptation. In the lifelong learning setting, a practical solution, as exemplified by TAIL (Liu et al., 2024b), involves initializing and training task-specific adapters for each task, subsequently freezing them, and retrieving the appropriate adapter based on the oracle task index. Such method allows efficient adaptation with only a small amount of task-specific parameters, but also exhibits certain limitations. First, assuming test-time task identity is known and retrieving adapter in a hard-coded way could be impractical in real scenarios. Second, failing to effectively share knowledge across isolated adapters result in poor forward transfer. Some prior arts construct a shared adapter pool and enable automatic timestep-level retrieval by, for example, performing query-key matching to select the one with highest similarity from a pool of adapters (Schmied et al., 2023), or using multifaceted prototypes to represent the cluster center in the state space for each adapter (Lee et al., 2024), but usually suffer from inaccurate retrieval and suboptimal performance.

In this paper, we introduce the Dynamic Mixture of Progressive Parameter-Efficient Expert Library (DMPEL) to address key challenges in lifelong robot learning. As illustrated in Figure 2, DMPEL features two core innovations: (1) it incrementally constructs a library of low-rank experts and employs a lightweight router to dynamically compose these experts into a unified policy based on the current context. This design allows the agent to flexibly adapt while leveraging all previously acquired parametric knowledge, thereby **enhancing forward transfer**; (2) it uses expert coefficient replay to regularize the router in accurately integrating experts for all previously encountered tasks, exploiting the modularity of fine-tuned parameters to **prevent catastrophic forgetting**. Notably, replaying low-dimensional embeddings and coefficients using the router is significantly more efficient than replaying demonstrations that require the entire policy in terms of storage and computation. Extensive evaluations on the lifelong manipulation benchmark LIBERO (Liu et al., 2024a) demonstrate that our method achieves superior forward transfer with reduced catastrophic forgetting compared to existing baselines, all while requiring minimal trainable parameters and storage.

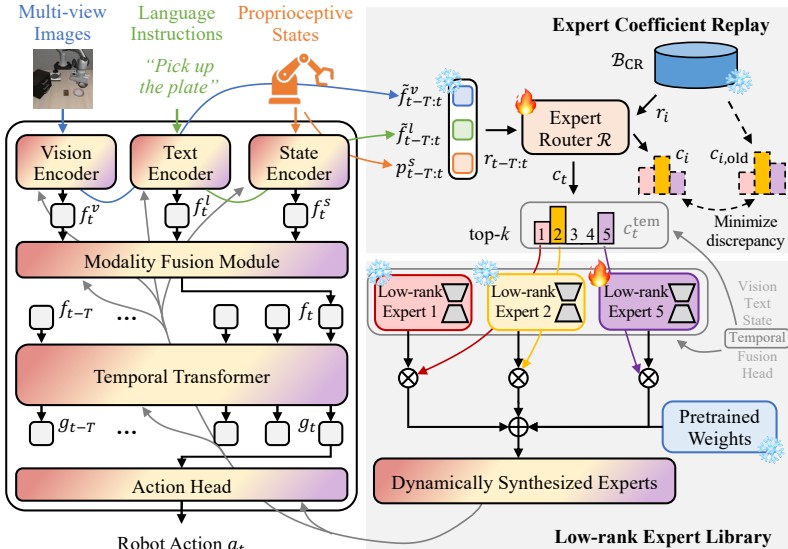

Figure 2: Overview of the proposed method DMPEL.

## 2 RELATED WORK

**Foundation Models for Robotics.** The traditional *tabula rasa* paradigm in robotics is widely criticized for its sample inefficiency, prompting researchers to explore the use of foundation models to improve downstream task transfer (Firoozi et al., 2023; Hu et al., 2023). One prominent direction involves transferring general visual representations: some studies directly employ foundation models from other domains as vision encoders (Parisi et al., 2022; Yuan et al., 2022), such as CLIP (Radford et al., 2021) and DINOv2 (Oquab et al., 2024), while others pretrain visual representations on egocentric video datasets (Nair et al., 2023; Majumdar et al., 2024). Another emerging line of research focuses on pretraining large-scale policies on diverse robotic tasks to capture generalizable behavior priors, for example, RT-2 (Brohan et al., 2023), OpenVLA (Kim et al., 2025), and $\pi_0$ (Black et al., 2024). However, when adapting these models to downstream tasks, most approaches either rely on the frozen pretrained model (i.e., zero-shot transfer) or require costly full fine-tuning. Some prior work has explored parameter-efficient fine-tuning (PEFT) by integrating task-specific modules into the pretrained vision encoder (Sharma et al., 2023; Marza et al., 2024) or temporal transformer (Liang et al., 2022; Xu et al., 2023; Qiao et al., 2023). Despite these advancements, the majority of these arts focus on single-task adaptation, whereas our approach targets lifelong adaptation to a sequence of new tasks.

**Lifelong Learning with PEFT.** Compared to full fine-tuning, PEFT optimizes only a small amount of inserted parameters while keeping the rest unchanged, which shares conceptual similarities with *architectural methods* in lifelong learning (Ding et al., 2023). Most studies on PEFT-based lifelong learning predominantly utilize prompt-based tuning (Jia et al., 2022; Lester et al., 2021) to adapt frozen pretrained model for tasks such as image classification, e.g., L2P (Wang et al., 2022c), DualPrompt (Wang et al., 2022b), CODA-Prompt (Smith et al., 2023), DAP (Jung et al., 2023), and text classification, e.g., EPI (Wang et al., 2023b), Progressive Prompts (Razdaibiedina et al., 2023). Current research primarily investigates strategies for dynamically selecting or generating instance-specific prompts (Zhou et al., 2024). Recognizing the representational limitations of prompt tuning, subsequent studies have extended to more powerful PEFT techniques, including LAE (Gao et al., 2023), HiDe-PET (Wang et al., 2025), and SKILL (Ge et al., 2024). Despite the success in incremental classification tasks, its application remains largely unexplored in robotics. A few pioneering works employ LoRA (Hu et al., 2021) to store task-specific knowledge, yet diverge in their retrieval mechanisms: TAIL (Liu et al., 2024b) relies on oracle task identifiers, L2M (Schmied et al., 2023) implements query-key matching between the current context and learnable keys, while IsCiL (Lee et al., 2024) establishes multifaceted prototypes through K-means clustering in the state space, with input states retrieving the closest LoRA parameters based on proximity. Our method incrementally constructs a parameter-efficient expert library and learns a lightweight router to dynamically retrieve and combine these experts.

**Mixture of Experts (MoE) and Model Fusion.** MoE approach involves training multiple specialized models for specific subtasks and has gained significant attention (Fedus et al., 2022; Muqeeth et al., 2024). A key method for combining these experts is parameter ensemble, supported by the linear mode connectivity phenomenon, which shows models converging to a single low-loss basin (Frankle et al., 2020). Task Arithmetic (Ilharco et al., 2023) introduces *task vectors*, representing the difference between fine-tuned and pretrained models, enhancing multi-task model performance when merged. Follow-up work (Yang et al., 2024; Tang et al., 2024) improves this by learning input-conditioned layer-wise fusion coefficients, increasing flexibility and performance. Model fusion techniques also apply to PEFT, where previously learned adapters are reused for efficient task adaptation (Wu et al., 2023; 2024; Wang et al., 2022a; Zhao et al., 2024), including LoRAHub (Huang et al., 2024) and AdapterFusion (Pfeiffer et al., 2021). In robotics, models like MELA (Yang et al., 2020), MoE-Loco (Huang et al., 2025), MORE (Zhao et al., 2025), and SDP (Wang et al., 2024b) create versatile, adaptive behaviors for multiple tasks by fusing expert sets. Our method aims to enhance forward transfer in lifelong robot learning by flexibly fusing previously learned low-rank experts for each sub-module in the policy according to the current context. Besides, we exploit the modularized design by replaying expert coefficient on the router to mitigate catastrophic forgetting.

## 3 PROBLEM FORMULATION

### 3.1 LIFELONG ROBOT LEARNING

In this paper, we aim to solve language-conditioned vision-based robot manipulation task, which can be formulated as a finite-horizon Markov Decision Process (MDP) $\mathcal{M} = (\mathcal{S}, \mathcal{A}, \mathcal{P}, \mathcal{H}, d_0, l)$, where $\mathcal{S}$ and $\mathcal{A}$ is the state space and action space of robot, $\mathcal{P} : \mathcal{S} \times \mathcal{A} \to \mathcal{S}$ is the transition dynamics, $\mathcal{H}$ is the episode length, and each task $\mathcal{T} \triangleq (d_0, l)$ is defined by the initial state distribution $d_0$ and the goal predicate in the form of language instruction $l : \mathcal{S} \to \{0, 1\}$. The ultimate objective of robot learning is to search for a policy $\pi$ that maximizes the expected success rate in reaching the goal: $\max_\pi J(\pi) = \mathbb{E}_{s_t, a_t \sim \pi, d_0} \left[ \sum_{t=1}^{\mathcal{H}} l(s_t) \right]$.

Due to the significant challenge of sparse reward in robot manipulation, we consider a more practical lifelong imitation learning setting (Liu et al., 2024a;b), where a robot sequentially learns over a stream of tasks $\{\mathcal{T}^1, \cdots, \mathcal{T}^K\}$ given an expert demonstration dataset $\mathcal{D}^k = \{\tau_n^k\}_{n=1}^N$ for each task $\mathcal{T}^k = (d_0^k, l^k)$, $1 \le k \le K$. Each expert trajectory $\tau_n^k$ can be represented as $(o_0, a_0, \cdots, o_\mathcal{H})$, in which observation $o_t$ at each step $t$ includes RGB images and proprioceptive states. Following common pratice in partially observable MDPs, we approximate the current state $s_t$ by stacking prior observations, i.e., $s_t = o_{\le t} = (o_0, \cdots, o_t)$. We perform behavior cloning (Bain & Sammut, 1995) to learn a stochastic policy $\pi_\theta$ by minimizing the negative log-likelihood loss $L$:

$$\min_\theta L(\theta) = -\mathbb{E}_{\tau_n^k \sim \mathcal{D}^k} \left[ \sum_{t=0}^{\mathcal{H}-1} \log \pi_\theta \left( a_t | o_{\le t}, l^k \right) \right]. \tag{1}$$

### 3.2 EVALUATION METRICS

We use three metrics to evaluate the performance of lifelong robot learning following prior arts (Liu et al., 2024a; Rodríguez et al., 2018): forward transfer (FWT), negative backward transfer (NBT), area under the success rate curve (AUC). Formally, we evaluate the policy on current task $\mathcal{T}^k$ when reaching a checkpoint $c \in \mathcal{C}$ (i.e., every multiple epochs of training) and we denote the success rate during evaluation as $S_{k,k,c}$, where the three subscripts refer to the current training task, the evaluated task, and the epoch number. When the training on current task $\mathcal{T}^k$ini is completed, we find the earliest checkpoint $c_k^*$ that achieves the best performance and mark it as the final success rate $S_{k,k}$, i.e., $S_{k,k} = S_{k,k,c_k^*} = \max_{c \in \mathcal{C}} S_{k,k,c}$. We only keep the best checkpoint for future training and evaluation as if the agent stops learning after $c_k^*$, i.e., for all $c \ge c_k^*$, $S_{k,k,c} = S_{k,k}$. This checkpoint $c_k^*$ will be further evaluated on previous tasks $\mathcal{T}^j$, $j \le k$, where the success rate is denoted as $S_{k,j}$. Intuitively, higher FWT indicates faster adaptation to new tasks, lower NBT indicates less catastrophic forgetting on old tasks, and higher AUC demonstrates an overall better performance across tasks and a good

balance between FWT and NBT. These metrics can be formally defined as:

$$\text{FWT} = \frac{1}{K} \sum_{k=1}^{K} \left[ \frac{1}{|\mathcal{C}|} \sum_{c \in \mathcal{C}} S_{k,k,c} \right], \tag{2}$$

$$\text{NBT} = \frac{1}{K} \sum_{k=1}^{K} \left[ \frac{1}{K-k} \sum_{l=k+1}^{K} (S_{k,k} - S_{l,k}) \right], \tag{3}$$

$$\text{AUC} = \frac{1}{K} \sum_{k=1}^{K} \left[ \frac{1}{K-k+1} \Big( \frac{1}{|\mathcal{C}|} \sum_{c \in \mathcal{C}} S_{k,k,c} + \sum_{l=k+1}^{K} S_{l,k} \Big) \right]. \tag{4}$$

## 4 PROPOSED METHOD

### 4.1 BASE POLICY ARCHITECTURE

We use a policy similar to the one used in the LIBERO benchmark (Liu et al., 2024a;b) and focus on developing better lifelong learning algorithms. The policy consists of vision, text, and state encoders, an input modality fusion module, a temporal transformer, and an action head.

**Vision, text, and state encoders.** The policy input is the historical observations $(o_{t-T}, \cdots, o_t)$ and the language instruction $l = \{l_i\}_{i=1}^{L}$, where $T$ is the context length and $L$ is the sentence length. The observation at each step $o_t = (I_t^1, \cdots, I_t^{N_c}, p_t)$ includes RGB images $I_t^{n_c}, 1 \le n_c \le N_c$ from $N_c$ different cameras and low-dimensional proprioceptive states $p_t$. In experiments, we use images from two cameras, the agent-view image and the eye-in-hand image. We employ the pretrained CLIP ViT-B/16 model (Radford et al., 2021) to serve as the vision encoder $\mathcal{E}_I$ and the text encoder $\mathcal{E}_L$. For the joint states and gripper states, we learn separate linear projection layers $\mathcal{E}_P$ to project the low-dimensional states to proprioceptive embeddings. In summary, at each timestep $t$, we obtain image embeddings $f_t^v$, text embeddings $f_t^l$, and proprioceptive embeddings $f_t^s$.

**Modality fusion.** We utilize a feature-wise linear modulation (FiLM) (Perez et al., 2018) to fuse language embeddings with the image and state embeddings. Generally, given the original feature $x$, the conditional input $z$, and the fusion network $f_{\text{FiLM}}$, we can obtain the modulated feature $x' = \gamma \odot x + \beta$ where $\gamma, \beta = f_{\text{FiLM}}(z)$. In our case, the conditional input $z$ refers to the language embedding $f_t^l$, $\gamma$ and $\beta$ are scaling and shifting vectors having the same size as $x$, and $\odot$ refers to element-wise multiplication.

**Temporal transformer and action head.** The causal temporal transformer backbone $\mathcal{D}_T$ processes a sequence of multi-modal embeddings to produce a latent vector $g_t$ at each decision-making timestep. We employ a Gaussian Mixture Model (GMM)-based output head $\mathcal{D}_H$ to compute the multi-modal distribution of manipulation actions (Liu et al., 2024b;a). During training, we optimize the negative log-likelihood loss between the action distribution and the ground truth action. At inference time, the robot executes the policy by using the mean of the Gaussian distribution with the highest density for end effectors.

### 4.2 LOW-RANK EXPERT LIBRARY

Following the pretrain-then-finetune paradigm, we pretrain the policy on a large-scale robotic dataset and freeze it during the lifelong adaptation to unseen downstream tasks. Low-Rank Adaptation (LoRA) (Hu et al., 2021) is a representative PEFT method that introduces learnable low-rank matrices $A \in \mathbb{R}^{d_{\text{in}} \times r}$, $B \in \mathbb{R}^{r \times d_{\text{out}}}$ and integrate in parallel with the frozen weight matrix $W_0 \in \mathbb{R}^{d_{\text{in}} \times d_{\text{out}}}$ through addition, i.e., $W_0 + AB$. By leveraging the low-rank decomposition $r \ll \min\{d_{\text{in}}, d_{\text{out}}\}$, LoRA substantially reduces the number of trainable parameters.

Existing methods usually learn a task-specific low-rank adapter $\Delta W_k = A_k B_k$ for each task $\mathcal{T}_k$ in lifelong robot learning (Liu et al., 2024b; Schmied et al., 2023). Such design requires task identifier during inference, lacks flexibility due to fixed capacity, and also hinders knowledge sharing across tasks. Prior arts in vision and language domain, e.g., LoRAHub (Huang et al., 2024) and

SD-LORA (Wu et al., 2025), use a task-wise learnable weight vector to combine existing low-rank experts for better performance on the new task. A key insight is that these low-rank experts captures complementary knowledge from a variety of specific tasks and can be leveraged to solve a new task. However, lifelong learning in robotic domain could be even more challenging. First, compared to image or text classification (De Lange et al., 2021; Masana et al., 2022; Wang et al., 2024a) that requires only single forward inference, robotic manipulation is a sequential decision-making process with long horizon. Besides, robot learning requires a mixed types of knowledge, for example, visual concepts, textual task goals, different spatial relationships, and successful adaptation to a new task requires a combination of adaptation in each sub-modules.

Following this motivation, we propose to learn a progressive low-rank expert library on top of the pretrained policy and dynamically reuse low-rank experts from prior tasks to facilitate efficient forward transfer to new task $\mathcal{T}_k$. We equip each chosen pretrained linear layer $\mathcal{F} = \{\boldsymbol{W}_0, \boldsymbol{b}_0\}$ with a low-rank expert library $\mathcal{L} = \{\boldsymbol{A}_j, \boldsymbol{B}_j, \boldsymbol{b}_j\}_{j=1}^k$ and use a lightweight router $\mathcal{R}$ to dynamically integrate pretrained and task-specific knowledge based on the current context, providing a more flexible solution for adaptation. Specifically, the router $\mathcal{R} : \mathbb{R}^{d_r} \to \mathbb{R}^{M \times k}$ is implemented as a multi-layer perceptron (MLP) to obtain the dynamic coefficient for each low-rank expert. The router takes the aggregated latent representation $\boldsymbol{r}_{t-T:t} = [\tilde{f}_{t-T:t}^v, \tilde{f}_{t-T:t}^l, p_{t-T:t}] \in \mathbb{R}^{d_r}$ as input, where the subscript $\cdot_{t-T:t}$ indicates mean pooling over the $T$-step context window. $\tilde{f}^v, \tilde{f}^l$ indicates the visual and text embeddings obtained from the frozen pretrained encoders, distinguishing them from $f^v, f^l$, which refer to the embeddings that have been modulated by the low-rank experts before being passed to the temporal transformer for action generation. The router input $\boldsymbol{r}_{t-T:t}$ is designed to comprehensively encode the current context while maintaining consistency across lifelong learning stages, which is then transformed into the coefficient vector for low-rank expert activation:

$$\boldsymbol{c}_t = \text{top-k}(\mathcal{R}(\boldsymbol{r}_{t-T:t}), \delta) \in \mathbb{R}^{M \times k}. \tag{5}$$

Note that the output $\boldsymbol{c_t}$ is divided into $M$ distinct expert coefficient vectors $\boldsymbol{c}_t^\diamond = [c_{1,t}^\diamond, c_{2,t}^\diamond, \cdots, c_{k,t}^\diamond] \in \mathbb{R}^k$, each assigned to a specific sub-module $\diamond$ in the policy. In our case, we have $M = 6$, consisting of vision encoder, text encoder, state encoder, modality fusion module, temporal transformer, and action head. This design allows each sub-module to use a different coefficient vector to integrate knowledge, providing larger flexibility in adaptation. The top-k$(\cdot, \delta)$ operator retains only the top $\delta$ over $k$ entries of the input vector at their original values, while setting all other coefficients to zero. This ensures efficient utilization of experts by reducing computational overhead while maintaining flexibility for adaptation. Finally, the input-conditioned linear layer $\{\tilde{\boldsymbol{W}}, \tilde{\boldsymbol{b}}\}$ is the dynamic mixture of the pretrained weight $\mathcal{F}$ and the low-rank experts from the library $\mathcal{L}$:

$$\boldsymbol{y} = \boldsymbol{x}\tilde{\boldsymbol{W}} + \tilde{\boldsymbol{b}} = \underbrace{\boldsymbol{x}\boldsymbol{W}_0 + \boldsymbol{b}_0}_{\text{pretrained}} + \underbrace{\boldsymbol{x}\sum_{j=1}^k c_{j,t}\boldsymbol{A}_j \sum_{j=1}^k c_{j,t}\boldsymbol{B}_j + \sum_{j=1}^k c_{j,t}\boldsymbol{b}_j}_{\text{fine-tuned}}. \tag{6}$$

After synthesizing the weights of all layers in the policy, the multi-modal historical observations are converted into robot actions in the manner described in Section 4.1. However, each sub-module within the policy has been dynamically modulated using the low-rank expert library applied on top of the pretrained weights.

### 4.3 EXPERT COEFFICIENT REPLAY

During lifelong adaptation, upon the completion of task $\mathcal{T}_k$, we freeze all low-rank experts in the library as acquired knowledge for future retrieval. Then we initialize a new learnable expert for task $\mathcal{T}_{k+1}$. In order to reduce the interference between tasks, we employ the Gram-Schmidt orthogonalization method on $\boldsymbol{A}_{k+1}$ (Smith et al., 2023; Wang et al., 2023a), while other elements $\boldsymbol{B}_{k+1}$ and $\boldsymbol{b}_{k+1}$ are simply zero-initialized. When training on the new task $\mathcal{T}_{k+1}$, the trainable components of the policy include the router $\mathcal{R}$ and the new LoRA experts $\boldsymbol{A}_{k+1}, \boldsymbol{B}_{k+1}, \boldsymbol{b}_{k+1}$. As the LoRA expert library expands, we also simultaneously increases the output dimension of the final linear layer of the router to keep them aligned. Our design allows the router either to directly reuse previous experts (by assigning zero magnitude on the new expert) or to rely on the new LoRA expert. We empirically investigate this in Section 5.2 on how to control the growing speed of the library and the router by pruning under-activated experts whose coefficients are always close to zero.

Although the previous LoRA experts are frozen, the router is continuously updated on sequentially arriving tasks. Therefore, incorrect integration of experts, i.e., the expert coefficients under the same context are inaccurate, may also lead to severe catastrophic forgetting and suboptimal performance.

Inspired by replay methods (Chaudhry et al., 2019; Xie & Finn, 2022; Zhao et al., 2024), we propose an expert coefficient replay mechanism to regularize the lightweight router on previously encountered tasks. The implementation involves two phases. During task finalization, we archive a subset of the router's input-output pairs, i.e., the context embedding $r$ and the coefficient vector $c$, in a dedicated buffer $\mathcal{B}_{\text{CR}}$. When adapting to new tasks, the core idea is to enforce consistency by requiring the router to generate coefficients that closely match the stored historical coefficients given the same context, which can be achieved through minimizing a mean squared error (MSE) objective:

$$L_{\text{CR}}(\mathcal{R}) = \mathbb{E}_{(r_i, c_{i,\text{old}}) \sim \mathcal{B}_{\text{CR}}} \frac{1}{2} \left( \mathcal{R}(r_i) - c_{i,\text{old}} \right)^2 . \tag{7}$$

With this approach, regardless of the number of tasks encountered, the coefficient vector stored in the buffer can be used to regularize the router. When faced with the same context, the router is encouraged to generate parameters $\{\tilde{W}, \tilde{b}\}$ similar to those originally produced during its training on previous tasks, thus exhibiting consistent behavior.

Compared to conventional experience replay methods that require storing entire trajectories and processing them through the entire policy, our expert coefficient replay strategy leverages the modularity of the low-rank expert library and focuses on regularizing the lightweight router using low-dimensional embeddings and coefficients, resulting in significant reductions in both computational overhead and storage requirements.

## 5 EXPERIMENTS

### 5.1 EXPERIMENTAL SETUP

We present an overview of the experimental setup below, with details included in Appendix A.

**Benchmark.** We conduct extensive evaluation in a lifelong robot learning benchmark LIBERO (Liu et al., 2024a). The robot is situated in a tabletop environment and equipped with a 6-DOF arm and a parallel gripper. We use four lifelong learning task suites: Goal, Spatial, Object, and Long. Each suite consists of 10 sequentially arriving robotic manipulation tasks designed to investigate the transfer of knowledge related to task goals, spatial information, and various objects. The benchmark also includes LIBERO-90, a collection of 90 short-horizon diverse tasks that serves as a pretraining dataset for downstream transfer. All tasks are consistent with the formulation in Section 3, i.e., generating continuous actions to control the robot based on multi-modal input.

**Baselines.** We consider sequential fine-tuning (SeqFT) baselines (Liu et al., 2024a) that either perform full fine-tuning (FFT) or use frozen pretrained feature (FPF) naively on sequentially arriving tasks. We also consider three representative lifelong learning methods that fine-tune all parameters, including Experience Replay (ER) (Chaudhry et al., 2019), Elastic Weight Consolidation (EWC) (Kirkpatrick et al., 2017), and PackNet (Mallya & Lazebnik, 2018). We also include a hierarchical baseline, LifelOng knowledge Transfer Using Skills (LOTUS) (Wan et al., 2024). Besides, we consider several LoRA-based methods that use different low-rank expert inserting and retrieving mechanism, including Task-specific Adapters for Imitation Learning (TAIL) (Liu et al., 2024b), Learning to Modulate (L2M) (Schmied et al., 2023), and Incremental Learning of Retrievable Skills for Continual Imitation Learning (IsCiL) (Lee et al., 2024). Since TAIL requires oracle task identifiers, we also provide the result of SeqFT (LoRA) to align with other methods. Unless otherwise specified, all lifelong methods begin with a policy pretrained (PT) on the LIBERO-90 dataset. We also include the multitask learning (MT) baseline, where the policy leans all tasks simultaneously. Since the AUC metric refers to the area under the success rate curve throughout the learning process, the final success rate of MT is generally regarded as an approximation of the upper bound (albeit not strictly) for any lifelong learning algorithm that are subject to catastrophic forgetting (Liu et al., 2024a).

**Implementation Details.** We utilize CLIP (Radford et al., 2021) as vision and text encoders, while all other components of the policy is learned from robot demonstrations. The LoRA rank is set to 8 for CLIP encoders and 16 for others, the same as in (Liu et al., 2024b). We pretrain the base policy on the LIBERO-90 dataset and use it as the starting point of lifelong learning. During adaptation,

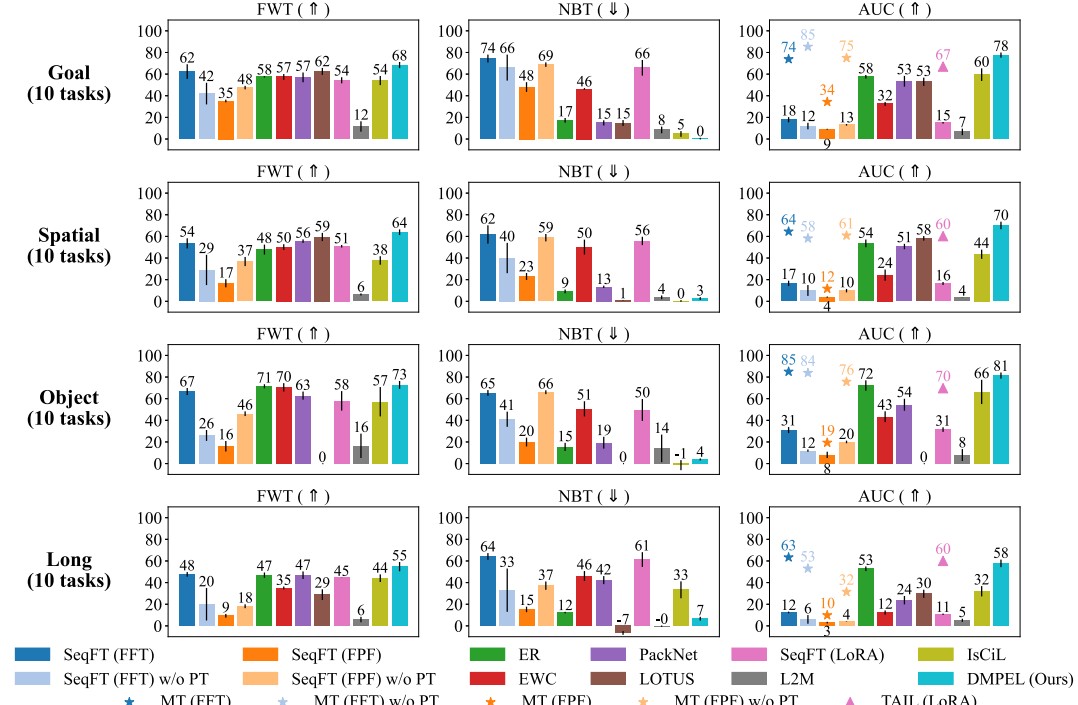

Figure 3: Performance of different lifelong robot learning methods on the LIBERO benchmark.

we learn 10 epochs on each arriving task and evaluate every 2 epochs, using a batch size of 32 and AdamW optimizer with a learning rate of 1e-4. We perform expert coefficient replay for 10 epochs after each task. As for the expert router, we employ a scaled sigmoid output activation to restrict the coefficient in [0,2] and use a top-3 strategy to select useful experts. All results are averaged over three random seeds.

## 5.2 RESULTS AND ANALYSIS

**Main Results.** Figure 4 presents the performance of the pretrained policy on both the original training tasks and zero-shot on unseen tasks, while Figure 3 summarizes three key metrics mentioned in Section 3.2 across all evaluated lifelong learning methods. Pretraining with FFT improves downstream transfer as expected, but still results in poor zero-shot performance, highlighting the need for more effective adaptation strategies. Besides, using frozen CLIP features is inefficient for robotic manipulation and performs even worse when pretraining is applied, likely due to overfitting in other sub-modules. Notably, there is a huge gap between multi-task (MT) and sequential fine-tuning (SeqFT). Among the full fine-tuning methods, ER achieves the best results, but this comes at the cost of storing and replaying large amounts of previous data (see Figure 1). LOTUS

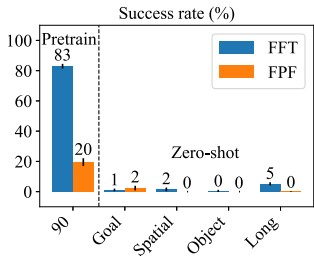

Figure 4: Performance of the pretrained policy on LIBERO

fails on the Object suite with the same code as on other suites, indicating poor robustness. Existing LoRA-based methods show lower forward transfer compared to FFT approaches, while DMPEL improves forward transfer by using dynamically synthesized experts, which reuse previous knowledge in a flexible way. Furthermore, our expert coefficient replay mechanism is highly effective in mitigating catastrophic forgetting, achieving near-zero NBT across all benchmarks without using oracle task identifiers. Conversely, L2M demonstrates weak forward transfer, likely due to its frozen action head. Although IsCiL reduces forgetting on simpler suites, it struggles with long-horizon tasks, presumably because of its fixed context encoding mechanism. In summary, DMPEL achieves efficient forward transfer and low catastrophic forgetting during lifelong learning, with only 1.2M trainable parameters (less than 0.7% of the policy) and minimal storage overhead.

**Ablation Studies.** In Figure 5a, we conduct an ablation study across two dimensions: the sparse activation of LoRA experts in the library (using top-1 expert, using top-3 experts, and soft merging

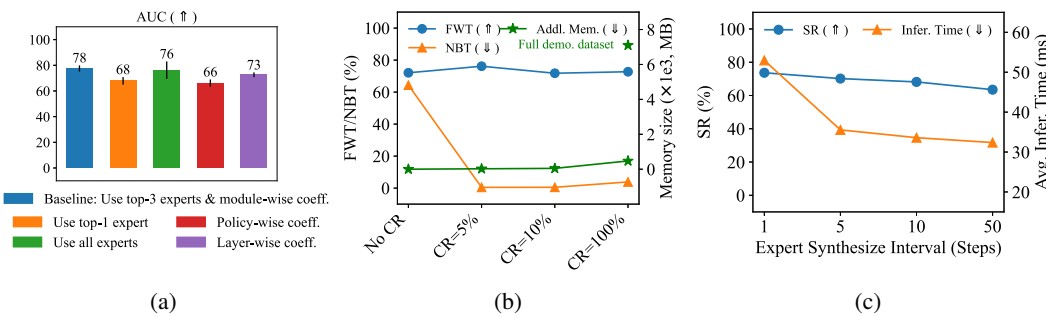

(a)        (b)        (c)

Figure 5: Ablation studies. (a) Sparse activation of LoRA experts and the granularity of expert coefficients on LIBERO-Goal (b) Different coefficient replay ratios on LIBERO-Object (c) Different expert synthesize intervals on LIBERO-Spatial.

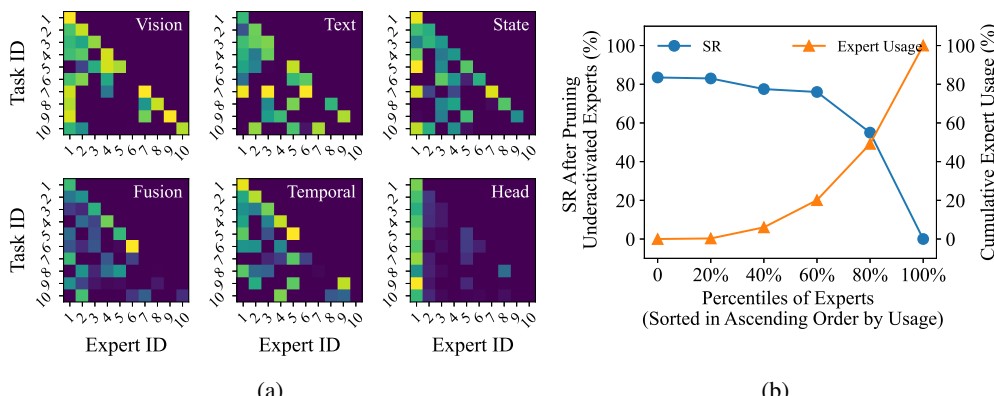

(a)              (b)

Figure 6: Analysis on expert coefficients on LIBERO-Object. (a) Visualization of the coefficients for each sub-module in the policy (indicated in the upper right corner), showing to what extent the expert is used by the current task and subsequently reused by later tasks during lifelong adaptation. (b) Influence on the average success rate (SR) when pruning underactivated experts.

all experts) and the granularity of expert coefficients (layer-wise, module-wise, and policy-wise). Utilizing only the top-3 experts yields better performance compared to using top-1 expert or all experts, demonstrating that appropriate sparsification is beneficial to both the efficiency and performance. Furthermore, assigning distinct coefficients to different submodules enables a more flexible and effective integration of knowledge while preserving simplicity. Figure 5b illustrates the impact of the coefficient replay (CR) ratio on overall performance. CR ratio refers to the proportion of router input-output pairs stored in the buffer for future replay. The results show that coefficient replay does not hinder forward transfer but helps mitigate forgetting efficiently, even when only a small fraction (e.g., 5%) of previous router input-output pairs is retained. Additionally, we demonstrate that replaying the context embeddings and coefficient vectors incurs a much lower storage overhead (approx. 6.7%) compared to replaying full expert demonstrations.

**Computational Overhead.** DMPEL introduces two additional stages: (1) encoding the current context with the frozen backbone and using the global router to compute routing coefficients, and (2) averaging the parameters of the top-$\delta$ experts to synthesize $\tilde{W} \in \mathbb{R}^{d_{in} \times d_{out}}$, which incurs approximately $2 \times \delta \times r \times (d_{in} + d_{out})$ FLOPs. The forward pass through the pretrained linear layer requires the usual $2 \times d_{in} \times d_{out}$ FLOPs. In robotic manipulation, which typically spans hundreds to thousands of decision steps and whose observations change only gradually from one step to the next. We investigate the influence of the weight synthesis interval (i.e., re-synthesizing policy parameters only every few steps during inference) on the success rate in LIBERO-Spatial, as shown in Figure 5c. The results indicate that the success rate experiences a slight decline as the granularity of the synthesis interval increases. Furthermore, as the synthesis interval expands from 1 to 5, the average inference time quickly approaches that of the backbone-only baseline (approximately 30 ms). This suggests two key points: (1) the expert router dynamically selects low-rank experts tailored to the current context, and less frequent synthesis may result in suboptimal expert coefficients; (2) we can

achieve a trade-off between the additional computational cost of parameter synthesis and the accuracy of expert coefficients, which directly impacts task performance.

**Visualization Analysis.** We present a visualization of the expert coefficients for each sub-module during the lifelong learning process on the LIBERO-Object benchmark. As illustrated in Figure 6a, significant knowledge sharing occurs across tasks, which contributes to the enhanced forward transfer capability of DMPEL. Notably, we observe that the low-rank expert learned for the action head in the first task is extensively reused in subsequent tasks, while the experts introduced later are rarely activated. This suggests that these inactive experts can be pruned to further improve storage efficiency without adversely affecting performance, as shown in Figure 6b. We also present t-SNE visualization in Figure 7, illustrating the latent embeddings from the temporal transformer's final block during the evaluation on Task 2. DMPEL maintains embedding consistency when adapting to new tasks, while other baselines demonstrate significant representation drift and catastrophic forgetting. Results from SeqFT with LoRA indicate that even a small number of parameters can substantially steer the pretrained model, and therefore we should be careful about the forgetting issue during lifelong PEFT. Both DMPEL and EWC impose constraints directly in the parameter space, ensuring consistent representation; however, EWC proves effective only in the short term (after Task 4) and fails to maintain effectiveness in the long run (after Task 8).

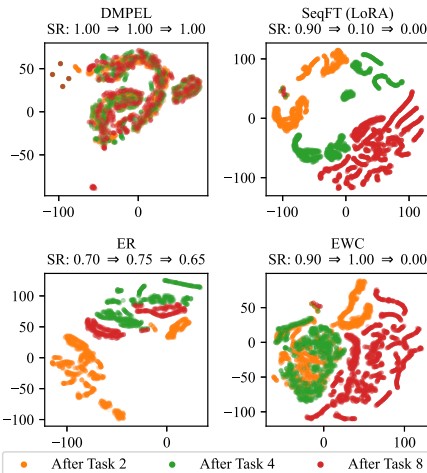

Figure 7: t-SNE visualization of embeddings from the final block when evaluated on Task 2 from LIBERO-Object.

**Additional Results.** Due to page limit, we provide additional results in Appendix B. Section B.1 presents further ablation studies on task order, number of demonstrations, rank size, router design, and top-k mismatch between training and evaluation. Section B.2 highlights the advantages of using a pretrained policy for lifelong learning compared to employing separate small models for each task. Section B.3 offers visualization analysis on the action space and expert activation trajectories. Sections B.4 and B.5 demonstrate the applicability of DMPEL in ultra-long task sequences and cross-domain adaptation, respectively. Finally, Section B.6 presents the learning curves of all methods.

## 6 CONCLUSION AND LIMITATION

We introduced Dynamic Mixture of Progressive Parameter-Efficient Experts Library (DMPEL) for lifelong robot learning. DMPEL incrementally builds a library of low-rank experts and employs a lightweight router to dynamically integrate them, enabling flexible adaptation across diverse scenarios. Leveraging policy modularity, we mitigate catastrophic forgetting via efficient coefficient replay on the router, facilitating expert retrieval without costly experience replay on the entire policy. Our extensive experiments on the LIBERO benchmark show that DMPEL surpasses state-of-the-art lifelong learning baselines in terms of forward transfer and catastrophic forgetting while requiring only very few trainable parameters and storage.

However, current work exhibits certain limitations. First, we conducted pretraining and adaptation experiments with a relatively small model, using only simulated data from the same robot. Future work will focus on scaling up to larger and more advanced policies, such as VLA models with billions of parameters, and exploring transferability to real robots or across different embodiments. In addition, we aim to further analyze the mechanisms of knowledge transfer and retention in lifelong learning algorithms, grounded in a solid theoretical framework. We provide further discussion on future directions in Appendix C.

## REPRODUCIBILITY STATEMENT

Our work is reproducible using the code we provide, which is currently available in the supplemental material and will be publicly released upon acceptance. Additionally, we include detailed pseudocode and hyperparameter settings in the Appendix.

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

## A  EXPERIMENTAL SETUP

### A.1  POLICY AND ALGORITHM DETAILS

**Policy Architecture.** We use a policy consisting of vision, text, and state encoders, an input modality fusion module, a temporal transformer, and an action head (Liu et al., 2024b;a). We utilize the open-source CLIP ViT/B-16 model (Radford et al., 2021) as vision and text encoders, while all other components of the policy is learned from robot demonstrations. Detailed hyperparameters of each sub-module are summarized in Table 1, most of them are naturally inherited from the open-source LIBERO benchmark and CLIP model.

**Baselines.** We consider naive sequential fine-tuning baselines, representative lifelong learning methods that involve tuning the entire policy, and several LoRA-based methods:

- Sequential Fine-tuning (SeqFT) (Liu et al., 2024a): Naively perform full fine-tuning (FFT) or use frozen pretrained feature (FPF), i.e., freeze the spatial encoders and fine-tune the rest, on sequentially arriving tasks.

- Experience Replay (ER) (Chaudhry et al., 2019), a *replay* method that maintains a memory buffer with samples from previous tasks. We store a subset of data after each task (20% in our experiments) and uniformly sample a fixed number of replay data from the buffer along with new task data during training. We use a batch size of 32 for old data, consistent with the batch size for new data, and mix them up for every training iteration.

- Elastic Weight Consolidation (EWC) (Kirkpatrick et al., 2017), a *regularization* method that uses the Fisher Information Matrix (FIM) to quantify the importance of each parameter and adds a regularization term to the learning objective to constrain network updates. We set the hyperparameters $\gamma = 0.9$ and $\lambda = 5 \times 10^4$.

- PackNet (Mallya & Lazebnik, 2018), an *architectural* method that first trains and prunes the network to retain a fixed proportion (25% in our experiments) of the most important parameters. The selected parameters are then fine-tuned and frozen to prevent forgetting.

- LifelOng knowledge Transfer Using Skills (LOTUS) (Wan et al., 2024) involves two stages: extracting temporal segment features by leveraging frozen foundation models for skill discovery, and hierarchical imitation learning with the meta-controller and a growing skill library. Since LOTUS employs a distinct architecture, we adjust its parameters to be comparable to our policy and follow the same training pipeline. Additionally, LOTUS utilizes the DINO-v2 (Oquab et al., 2024) foundation model instead of CLIP, as recommended by the ablation study in their paper.

- Task-specific Adapters for Imitation Learning (TAIL) (Liu et al., 2024b) introduces separate LoRAs into the spatial encoder and temporal decoder for each new task. The fusion module, state encoder, and action head are also full fine-tuned and stored as task-specific adapters. TAIL requires an oracle task identifier to retrieve the corresponding adapters during inference. In order to align with other methods, we additionally provide the result of SeqFT (LoRA), which fine-tunes LoRA parameters sequentially.

- Learning to Modulate (L2M) (Schmied et al., 2023) maintains a learnable modulation pool with keys and associated modulators. The embedded history of state tokens serves as the query vector for query-key matching, which finally steers the pretrained policy's behavior. We follow the original paper to use a frozen fusion module and action head, and insert LoRA pools in the CLIP encoders and the temporal transformer backbone. We use a pool size of 30 and the similarity loss coefficient $\lambda = 0.5$.

- Incremental Learning of Retrievable Skills for Continual Imitation Learning (IsCiL) (Lee et al., 2024) establishes multifaceted prototypes through K-means clustering in the state space, where proper LoRA-based skills is retrieved upon current input state. We define the task-specific skill the same as in TAIL (Liu et al., 2024b), i.e., LoRA matrices for transformer, the fusion module, the state encoder, and the action head. We use CLIP encoded features as context embeddings and set the skill prototype $\mathcal{X}_z$ in $\mathcal{X}$ to be composed of 20 bases.

We utilize the implementations of SeqFT, ER, EWC, and PackNet provided in LIBERO (Liu et al., 2024a) and we directly use the official code of LOTUS (Wan et al., 2024). When training with ER

Table 1: Policy Configurations

| Hyperparameter | Value |
|---|---|
| *Vision & Text Encoder* | |
| Pretrained model | CLIP ViT-B/16 |
| Image resolution | 128*128 |
| *Proprioceptive States* | |
| Joint state dim. | 7 |
| Gripper state dim. | 2 |
| *Modality Fusion* | |
| Number of hidden layers | 1 |
| Hidden size | 256 |
| Activation | GELU |
| *Temporal Transformer* | |
| Number of layers | 6 |
| Number of attn. heads | 8 |
| Embedding size | 768 |
| MLP hidden size | 1024 |
| Max sequence Length | 10 |
| Dropout | 0.15 |
| *Action Head* | |
| Number of hidden layers | 2 |
| Hidden size | 256 |
| Number of Modes | 5 |
| Min. std | 1e-4 |
| Activation | Softplus |
| Action dim. | 7 |
| *Expert router* | |
| Number of hidden layers | 2 |
| Hidden Activation | GELU |
| Output Activation | Sigmoid |

Table 2: Training Hyperparameters

| Hyperparameter | Value |
|---|---|
| *Optimizer* | |
| Training epochs | 10 |
| Batch size | 32 |
| Optimizer | AdamW |
| Betas | (0.9,0.999) |
| Learning rate | 1e-4 |
| Anneal strategy | cos |
| Weight decay | 0.1 |
| Gradient clip | 100 |
| *Image Augmentation* | |
| Brightness | 0.3 |
| Saturation | 0.3 |
| Contrast | 0.3 |
| Hue | 0.3 |
| Color jitter prob. | 0.9 |
| Rotation | 15 |
| Translation | 0.1 |
| Affine prob. | 0.6 |
| *Evaluation* | |
| Epochs per eval. | 2 |
| Eval. episodes | 20 |

and EWC, we use two GPUs, each with a batch size of 16, ensuring the total batch size remains consistent with other methods. Due to the unavailability of publicly released code for TAIL, we implemented it based on the description in the paper (Liu et al., 2024b). We integrated the code for L2M and IsCiL into our codebase, as the original implementations operate with different policies in different benchmarks.

For LoRA-based methods, we insert LoRA modules with rank $r = 8$ into the 6-th to 11-th layers of the CLIP image and text encoders, and LoRA modules with $r = 16$ into the temporal transformer. LoRA is applied to the query and value projection matrices. In TAIL and IsCiL, we learn task-specific fusion modules, state encoders, and action heads, as described in (Liu et al., 2024b). In contrast, these components are frozen in L2M following the original paper (Schmied et al., 2023).

**DMPEL (Ours).** We adopt the same settings in the image encoder, text encoder, and temporal transformer as other LoRA-based methods, and use LoRA experts with $r = 16$ for all linear layers in other sub-modules. Besides, in the Long suite, we tune both the low-rank weights $A, B$ and biases $b$, whereas in the Goal, Spatial, and Object suites, we only tune the low-rank weights $A, B$. However, both settings lead to fewer learnable parameters than TAIL (Liu et al., 2024b), which learns these sub-modules separately for each task. We also employ dropout with probability $p = 0.15$ on the coefficient for low-rank experts to avoid overfitting.

We provide a graphic overview of DMPEL in Figure 2 in the main text and the detailed pseudocode in Algorithm 1 (Training) and 2 (Inference). For further implementation specifics, please refer to the source code.

---

**Algorithm 1** Training Process of DMPEL

---

**Require:** Pretrained policy $\pi_\theta$, router $\mathcal{R}$, low-rank expert library $\mathcal{L} = \varnothing$ for each layer $\mathcal{F}$, coefficient replay buffer $\mathcal{B}_{\text{CR}} = \varnothing$

1: **for** $k = 1, 2, \cdots, K$ **do**
2:     **for** each $\{\mathcal{F}, \mathcal{L}\}$ **do**
3:         Initialize $\boldsymbol{A}_k, \boldsymbol{B}_k$, (and optionally $\boldsymbol{b}_k$)
4:         $\mathcal{L} \leftarrow \text{sg}(\mathcal{L}) \cup \{\boldsymbol{A}_k, \boldsymbol{B}_k, \boldsymbol{b}_k\}$ # sg($\cdot$) means stop gradient, i.e., freezing previously learned low-rank experts
5:     **end for**
6:     **for** epochs $c = 1, 2 \cdots, C$ **do**
7:         **for** each mini-batch data from dataset $\mathcal{D}^k$ **do**
8:             Compute the context embedding $\boldsymbol{r}$ using the frozen encoders in the pretrained policy $\pi_\theta$
9:             Compute the expert coefficient vector $\boldsymbol{c}$ with the router $\mathcal{R}$ using Eq. (5)
10:             **for** each $\{\mathcal{F}, \mathcal{L}\}$ **do**
11:                 Obtain the expert coefficient $\boldsymbol{c}^\diamond$ according to which sub-module $\diamond$ the layer belongs to
12:                 Synthesize parameters $\tilde{\boldsymbol{W}}, \tilde{\boldsymbol{b}}$ using Eq. (6)
13:             **end for**
14:             Compute behavior cloning loss $L$ using Eq. (1)
15:             Update router parameters $\mathcal{R}$ and low-rank expert parameters $\boldsymbol{A}_k, \boldsymbol{B}_k$, (and optionally $\boldsymbol{b}_k$) to minimize $L$
16:             **if** $c = C$ **then** $\mathcal{B}_{\text{CR}} \leftarrow \mathcal{B}_{\text{CR}} \cup \{(\boldsymbol{r}, \boldsymbol{c})\}$ with a probability equals to the coefficient replay ratio
17:         **end for**
18:         **if** $c \in \mathcal{C}$ **then** Evaluate on task $\mathcal{T}^k$ to obtain success rate $s_{k,k,c}$ (Algorithm 2)
19:     **end for**
20:     **for** epochs $c = 1, 2 \cdots, C$ **do**
21:         **for** each mini-batch data from the buffer $\mathcal{B}_{\text{CR}}$ **do**
22:             Compute expert coefficient replay loss $L_{\text{CR}}$ using Eq. (7)
23:             Update router parameters $\mathcal{R}$ to minimize $L_{\text{CR}}$
24:         **end for**
25:     **end for**
26:     **for** $j = 1, 2, \cdots, k$ **do**
27:         Evaluate on task $\mathcal{T}^j$ to obtain success rate $s_{k,j}$ (Algorithm 2)
28:     **end for**
29: **end for**
30: Compute metrics FWT, NBT, AUC using Eq. (2)-(4)

---

## A.2 TRAINING CONFIGURATION

We generally follow the setup in the LIBERO benchmark for training and evaluation (Liu et al., 2024a). For all experiments, we use either Nvidia A100 or H800 GPUs. We employ Distributed Data Parallel (DDP) for parallel training across 4 GPUs during pretraining, while using a single GPU for lifelong learning. We employ 16-bit floating point precision (FP16) for training and inference to accelerate the process, with the exception of the router, which uses FP32. This selective precision technique is also discussed in (Fedus et al., 2022). To ensure reproducibility and statistical significance, we adopted three random seeds (100, 200, 300) and reported the mean and standard deviation of the performance over three seeds throughout the paper.

We deviate from the original LIBERO benchmark by training each task for only 10 epochs (evaluating every 2 epochs), instead of 50 epochs (evaluating every 10 epochs), as we observed convergence typically within 10 epochs. Consequently, metrics including FWT and AUC that are averaged over epochs generally appear lower than those reported in the original LIBERO paper and some follow-up work. We also explored various hyperparameter settings for DMPEL, such as top-k selection, replay ratio coefficients, and weight initialization; these ablations are detailed in Figure 5. Otherwise, we use the same training hyperparameters across all methods and benchmarks, which proved effective. Hyperparameter details are summarized in Table 2.

---

**Algorithm 2** Inference Process of DMPEL

---

**Require:** Pretrained policy $\pi_\theta$, router $\mathcal{R}$, each layer with pretrained weight and low-rank expert library $\{\mathcal{F}, \mathcal{L}\}$, FIFO queues $F_q = \varnothing$, $R_q = \varnothing$ with maximum length $T$

1: **for** $t = 1, 2, \cdots, \mathcal{H}$ **do**
2:     Observe $o_t = (I_t^1, \cdots, I_t^{N_c}, p_t)$ and language instruction $l$
3:     **if** $T_{\mathrm{syn}} \mid t$ **then**
4:         Compute visual and textual embeddings $\tilde{f}_t^v, \tilde{f}_t^l$ using the frozen CLIP encoders $\mathcal{E}_I, \mathcal{E}_L$ ($\mathcal{F}$ only)
5:         Compute the context embedding $r_t = [\tilde{f}_t^v, \tilde{f}_t^l, p_t]$
6:         $R_q \leftarrow R_q \cup \{r_t\}$
7:         Compute the expert coefficient vector $c_t = \text{top-k}(\mathcal{R}(r_{t-T:t}), \delta)$
8:         **for** each $\{\mathcal{F}, \mathcal{L}\}$ **do**
9:             Obtain the expert coefficient $c_t^\diamond$ according to which sub-module $\diamond$ the layer belongs to
10:             Synthesize parameters $\tilde{W}, \tilde{b}$ using Eq. (6)
11:         **end for**
12:     **else**
13:         Reuse previous visual and text embeddings $\tilde{f}_t^v \leftarrow \tilde{f}_{t-1}^v$, $\tilde{f}_t^l \leftarrow \tilde{f}_{t-1}^l$
14:         $R_q \leftarrow R_q \cup \{[\tilde{f}_t^v, \tilde{f}_t^l, p_t]\}$
15:     **end if**
16:     Compute visual, textual, and state embeddings $f_t^v, f_t^l, f_t^s$ using encoders $\mathcal{E}_I, \mathcal{E}_L, \mathcal{E}_P$ ($\mathcal{F}$ and $\mathcal{L}$)
17:     Obtain $f_t \in \mathbb{R}^{\text{num of modalities} \times \text{embedding dim}}$ using FiLM to fuse language embeddings with image and state embeddings
18:     $F_q \leftarrow F_q \cup \{f_t\}$
19:     Compute latent embedding $g_t = \mathcal{D}_T(f_{t-T}, \cdots, f_{t-1}, f_t)$
20:     Predict action $a_t = \mathcal{D}_H(g_t)$
21:     Interact with the environment using action $a_t$ and obtain done flag $d_{t+1}$
22:     **if** $d_{t+1}$ is True **then break**
23: **end for**

---

### A.3 BENCHMARK DETAILS

We present the specific language instructions for the tasks in four lifelong learning LIBERO task suites in Table 3. Although some tasks have similar descriptions, they are not identical due to differences in environment configurations, such as varying spatial layouts, objects, or goal positions. Each task provides 50 successful expert demonstrations for imitation learning.

## B ADDITIONAL RESULTS

### B.1 ABLATION STUDIES

**Task Order.** We further investigate the performance of DMPEL across various settings. First, we assess its robustness to different task orders, as shown in Figure 8a. The default order provided by the LIBERO benchmark is 0123456789, the reverse order is 9876543210, and the randomly generated task order is 4687312095.

**Number of Demos.** We illustrate the impact of the number of demonstrations for each new task in Figure 8c. The performance of FWT remains quite stable when the number of demonstrations ranges from 10 to 50, indicating that DMPEL can effectively adapt across tasks even in low-data regimes.

**Rank Size.** We analyze the impact of the rank size of each LoRA expert in Figures 8b. Performance generally improves as the rank size increases, but shows saturation at a rank size of 32, indicating a trade-off between parameter efficiency and performance. Additionally, we compare the lifelong

Table 3: Tasks Instructions of LIBERO Task Suites

| Benchmark Suite | Task Instructions |
|---|---|
| LIBERO-Goal | open the middle drawer of the cabinet
put the bowl on the stove
put the wine bottle on top of the cabinet
open the top drawer and put the bowl inside
put the bowl on top of the cabinet
push the plate to the front of the stove
put the cream cheese in the bowl
turn on the stove
put the bowl on the plate
put the wine bottle on the rack |
| LIBERO-Spatial | pick up the black bowl between the plate and the ramekin and place it on the plate
pick up the black bowl next to the ramekin and place it on the plate
pick up the black bowl from table center and place it on the plate
pick up the black bowl on the cookie box and place it on the plate
pick up the black bowl in the top drawer of the wooden cabinet and place it on the plate
pick up the black bowl on the ramekin and place it on the plate
pick up the black bowl next to the cookie box and place it on the plate
pick up the black bowl on the stove and place it on the plate
pick up the black bowl next to the plate and place it on the plate
pick up the black bowl on the wooden cabinet and place it on the plate |
| LIBERO-Object | pick up the alphabet soup and place it in the basket
pick up the cream cheese and place it in the basket
pick up the salad dressing and place it in the basket
pick up the bbq sauce and place it in the basket
pick up the ketchup and place it in the basket
pick up the tomato sauce and place it in the basket
pick up the butter and place it in the basket
pick up the milk and place it in the basket
pick up the chocolate pudding and place it in the basket
pick up the orange juice and place it in the basket |
| LIBERO-Long | put both the alphabet soup and the tomato sauce in the basket
put both the cream cheese box and the butter in the basket
turn on the stove and put the moka pot on it
put the black bowl in the bottom drawer of the cabinet and close it
put the white mug on the left plate and put the yellow and white mug on the right plate
pick up the book and place it in the back compartment of the caddy
put the white mug on the plate and put the chocolate pudding to the right of the plate
put both the alphabet soup and the cream cheese box in the basket
put both moka pots on the stove
put the yellow and white mug in the microwave and close it |

PEFT baselines (TAIL, IsCiL, and DMPEL) under the same total number of LoRA experts. Notably, DMPEL with top-3 activation activates three LoRA experts simultaneously, while the others activate only one. Therefore, we present the results for TAIL and IsCiL with the rank increased by three times in Table 4. However, the results show that naively increasing the rank size for the baselines does not achieve the same performance as DMPEL, highlighting the effectiveness of using a LoRA expert library with expert coefficient replay.

**Router Design.** We utilize the Sigmoid function to constrain the router output instead of applying Softmax normalization. As noted in the related work, DMPEL is heavily inspired by research on model fusion and merging, such as SD-LoRA (Wu et al., 2025) and LoRAHub (Huang et al., 2024),

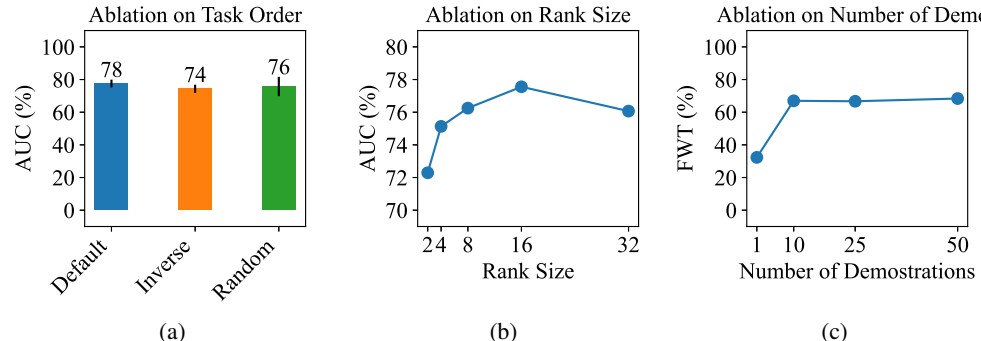

Figure 8: Ablation studies in the LIBERO-Goal suite. (a) Different task orders (b) Different rank sizes (c) Different number of demonstrations.

which demonstrate that adjusting only the magnitude of LoRA can yield promising transfer results. The interval [0, 2] serves as a hyperparameter based on intuition to limit the magnitude, with 1 as the midpoint, representing the direct use of the previous LoRA module without magnitude alteration. We empirically compare the performance of Sigmoid and Softmax with various top-k strategies, as shown in Figure 9. Notably, router with Sigmoid generally outperforms the ones with Softmax, and the top-1 with Softmax variant exhibited the lowest performance. We hypothesize that using raw coefficients to modulate the magnitudes of LoRA experts provides greater flexibility, as these coefficients are independent for each expert. In contrast, Softmax normalization enforces that the sum of coefficients equals 1, meaning that increasing one coefficient necessarily decreases another. During expert coefficient replay, we apply the MSE loss between old and new coefficients in all cases for its simplicity. When transitioning from task $k$ to task $k + 1$, we pad the coefficient vector with 0 to maintain alignment with the current library size and ensure that later introduced experts are excluded from previous tasks. We hypothesize that minimizing the discrepancy of logits may be a more effective approach for the top-1 Softmax variant due to its one-hot property. In this context, padding zeros to the coefficients becomes padding $-\infty$ to the logits. We leave further investigation of this for future work.

**Top-k Mismatch between Training and Evaluation.** We present the performance of DMPEL with a differing number of top-k LoRA experts between training and evaluation in Table 5. Although there is only a slight decrease in success rate (e.g., 2%), this allows for adaptive scaling based on computational resource constraints. Furthermore, training with multiple experts while using the top-1 expert yields better performance than training with only the top-1 expert, likely due to enhanced robustness during training.

## B.2 PRETRAINED POLICY WITH PEFT VS SEPARATE POLICY

The *pretrain-then-finetune* paradigm has proven to be highly successful in the domains of vision, language, and robotics. It is widely believed that pretrained models capture common knowledge that can be leveraged for various downstream tasks, enabling rapid transfer or even zero-shot transfer. Meanwhile, *parameter-efficient fine-tuning* (PEFT) techniques have shown great effectiveness in steering frozen foundation models by incorporating small, learnable modules during adaptation.

Table 4: Performance on LIBERO-Goal with Different Rank Size

| Method | Rank Size | FWT | BWT | AUC |
|---|---|---|---|---|
| TAIL (with task ID) | rank | $0.54 \pm 0.03$ | 0 | $0.67 \pm 0.06$ |
| TAIL (with task ID) | rank $\times$ 3 | $0.55 \pm 0.03$ | 0 | $0.71 \pm 0.02$ |
| ISCiL | rank | $0.54 \pm 0.04$ | $0.05 \pm 0.03$ | $0.60 \pm 0.06$ |
| ISCiL | rank $\times$ 3 | $0.52 \pm 0.02$ | $0.09 \pm 0.06$ | $0.59 \pm 0.05$ |
| DMPEL w/ top-1 | rank | $0.61 \pm 0.03$ | $0.01 \pm 0.02$ | $0.68 \pm 0.03$ |
| DMPEL w/ top-3 | rank | $0.68 \pm 0.03$ | $0.00 \pm 0.01$ | $0.78 \pm 0.02$ |

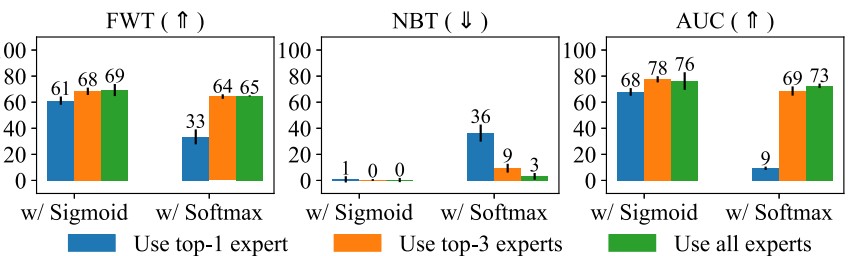

Figure 9: Ablation studies on router design in the LIBERO-Goal suite.

Therefore, the results presented in the main text are based on a setting where a large policy is first pretrained on the LIBERO-90 dataset before being transferred to streaming tasks. We also demonstrate the performance gains achieved by pretraining compared to starting from scratch. To further illustrate effectiveness, we compare our approach with training separate models for each individual task in Table 6. We present results using the policy ViT-T provided in the original LIBERO benchmark with two different sizes: the smaller model has 1.9M parameters (similar to that of a LoRA expert), while the larger model has 17.7M parameters (the total number of all models after 10 tasks will be comparable to that of the single large policy used in the main experiment). The results reveal a significant gap between using dedicated small models and employing a single large pretrained policy enhanced with LoRA experts, underscoring the effectiveness of DMPEL.

### B.3 VISUALIZATION ANALYSIS

**LoRA Expert Activation.** We plot some representative trajectories from different benchmark suites in Figures 11-13, with labels indicating the expert ID from the library. In Figure 11, we observe that the policy relies on the newly introduced expert 2 when executing task 2. In task 9 that also involves picking up a bowl but placing it down at a different destination, the vision encoder consistently activates expert 2 throughout the trajectory, while the temporal decoder primarily reuses expert 2 during the first half of the trajectory, likely due to a similar action pattern. Similarly, in Figure 12, we also see that the LoRA expert 4 is actively reused by the vision and state encoder in task 7. Additionally, as the robot executes task 7, we can see the switching process between different experts while approaching, grasping, transporting, and placing down the bowl. In Figure 13, the vision encoder reuses previously learned expert but the fusion sub-module learns a new expert to effectively fuse features for manipulation.

**Action Space Coverage.** In Section 5.2 Figure 6, we empirically observe that the low-rank expert learned for the action head in the first task is extensively reused in subsequent tasks, while the experts introduced later are rarely activated. The LIBERO benchmark is built on the Robosuite framework (Zhu et al., 2020), where the default underlying controller is OSC_POSE (Operational Space Control with Pose). The action includes the 6D end-effector (EEF) poses (position & orientation) and the status of gripper, $a = [\mathrm{d}x, \mathrm{d}y, \mathrm{d}z, \mathrm{d}\alpha, \mathrm{d}\beta, \mathrm{d}\gamma, gripper]$. Seven dimensions respectively corresponds to translation along the $x$-axis, $y$-axis, and $z$-axis, the roll ($\alpha$), pitch ($\beta$), and yaw ($\gamma$) rotation, along with the open/close status of the gripper. We visualize ten action trajectories from the LIBERO-Object benchmark in Figure 10, illustrating that coverage of the action space varies from task to task. Prior research has demonstrated that using different action spaces (e.g., position control vs. velocity

Table 5: Performance on LIBERO-Goal with Different Top-k Experts Used in Training and Evaluation

| Training | Evaluation | Average Success Rate |
|---|---|---|
| Use all experts | Use all experts | $0.79 \pm 0.06$ |
| Use all experts | Use top-3 experts | $0.81 \pm 0.05$ |
| Use all experts | Use top-1 experts | $0.77 \pm 0.05$ |
| Use top-3 experts | Use top-3 experts | $0.81 \pm 0.05$ |
| Use top-3 experts | Use top-1 experts | $0.79 \pm 0.01$ |
| Use top-1 expert | Use top-1 expert | $0.69 \pm 0.05$ |

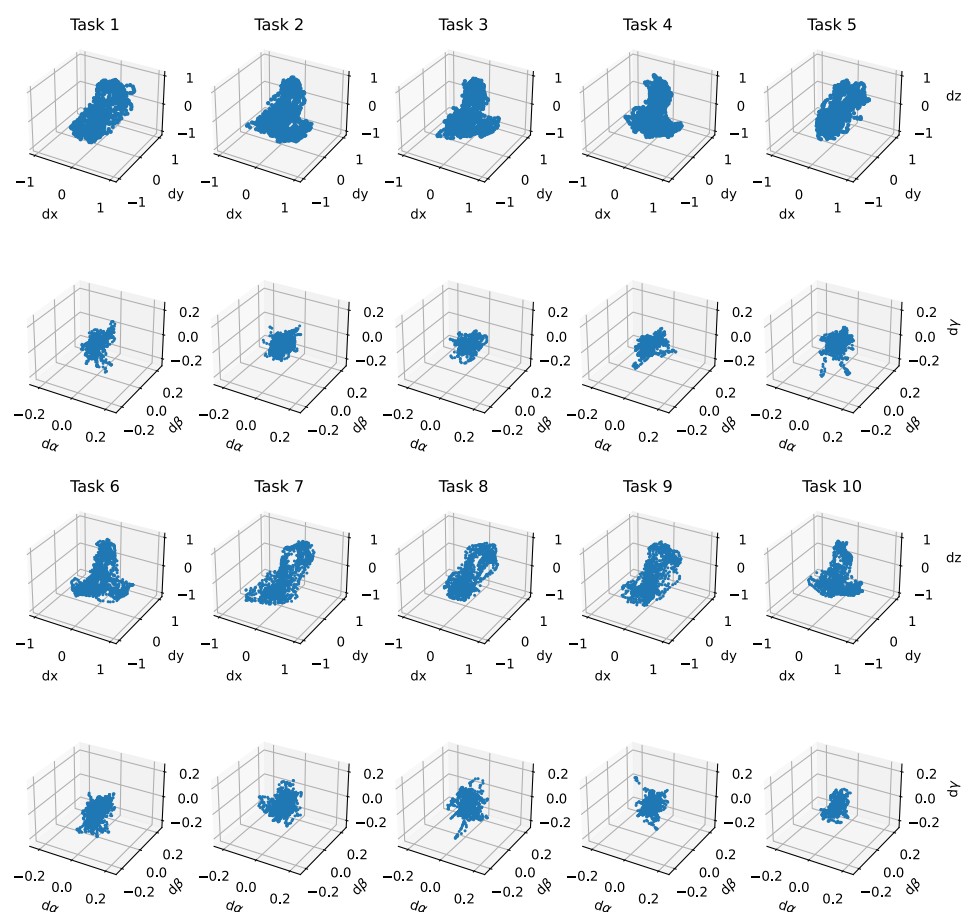

Figure 10: Visualization of the action trajectories from demonstrations of LIBERO-Object benchmark.

control) can significantly affect control performance (Zhao et al., 2023; Chi et al., 2025), and we intend to explore this further in future work.

### B.4 ULTRA-LONG TASK SEQUENCE

Beyond the experiment on the standard $K = 10$ task sequence form the LIBERO benchmark, we also investigate the applicability of DMPEL to ultra-long sequences. To this end, we concatenate the LIBERO-Object, Goal, and Spatial benchmark to form a task sequence with $K = 30$. Following the empirical analysis in Section 5.2 and illustrated in Figure 6b, we prune underactivated low-rank experts every 10 tasks to control the linear growth of the library. At the end of the 30-th task, as shown in Table 7, the number of LoRA experts in each module (the elements of the six-dimensional vector in order representing the vision encoder, text encoder, state encoder, modality fusion module, temporal transformer, and action head) typically remains below 30, resulting in a compact library that exhibits effective knowledge sharing. The results shown in Figure 14 and Table 7 indicate that DMPEL consistently outperforms FFT with experience replay in success rate, while incurring significantly

Table 6: Performance on LIBERO-Goal with a Separate Policy for each Individual Task

| Method | Number of Parameters | FWT | BWT | AUC |
|---|---|---|---|---|
| ViT-T (Small) | 1.9M ($\times 10$) | $0.04 \pm 0.01$ | 0 | $0.08 \pm 0.01$ |
| ViT-T (Large) | 17.7M ($\times 10$) | $0.21 \pm 0.04$ | 0 | $0.43 \pm 0.04$ |
| DMPEL | 174.1M + 1.2M ($\times 10$) | $0.68 \pm 0.03$ | $0.00 \pm 0.01$ | $0.78 \pm 0.02$ |

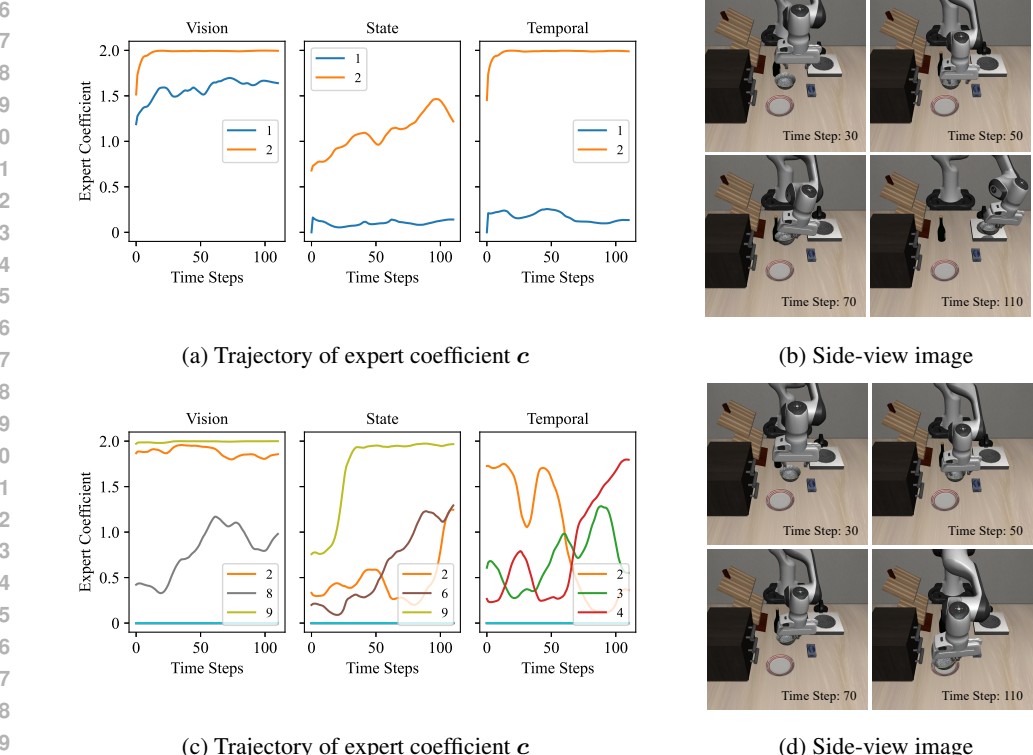

(a) Trajectory of expert coefficient $c$

(b) Side-view image

(c) Trajectory of expert coefficient $c$

(d) Side-view image

Figure 11: Visualization Analysis on Expert Activation: (a)-(b) Task 2 from LIBERO-Goal: put the bowl on the stove; (c)-(d) Task 9 from LIBERO-Goal: put the bowl on the plate.

lower computational and storage costs. However, we still observe a gradual decline in performance as the number of tasks increases. Given the small size of the router, this is not surprising. When the number of distinct tasks continue to increase, the plasticity of the router may be exhausted. One potential solution is to progressively scale up the router as the task count increases. Since all expert coefficients are stored in the replay buffer, we should be able to recover the coefficient distribution in the enlarged router. We plan to explore this in future research.

### B.5 CROSS-DOMAIN ADAPTATION

While LIBERO is a large-scale multi-modal benchmark that requires complex behavior and hence giving promising results, we further evaluate the cross-domain adaptation capabilities of DMPEL. We conduct experiments with the same hyperparameters on three tasks (Lift, Can, and Square) from Robomimic (Mandlekar et al., 2022), which together form a task sequence with $K = 3$. These tasks share the same observation and action space, but feature novel backgrounds and objects. The results presented in Table 8 demonstrate that DMPEL can be effectively applied for lifelong adaptation in the Robomimic benchmark.

Table 7: Storage and Computational Cost at the End of an Ultra-long Task Sequence ($K = 30$)

| Method | Number of LoRA Experts and Parameters in Each Module | Trainable Parameters | Additional Storage for Replay |
|---|---|---|---|
| DMPEL | [15,18,17,21,17,10], 10.7M (seed=100)
[18,19,17,19,17,7], 11.1M (seed=200)
[20,18,20,19,18,9], 11.7M (seed=300) | 1.2M | 0.06GB |
| ER | / | 174.1M | 3.7GB |

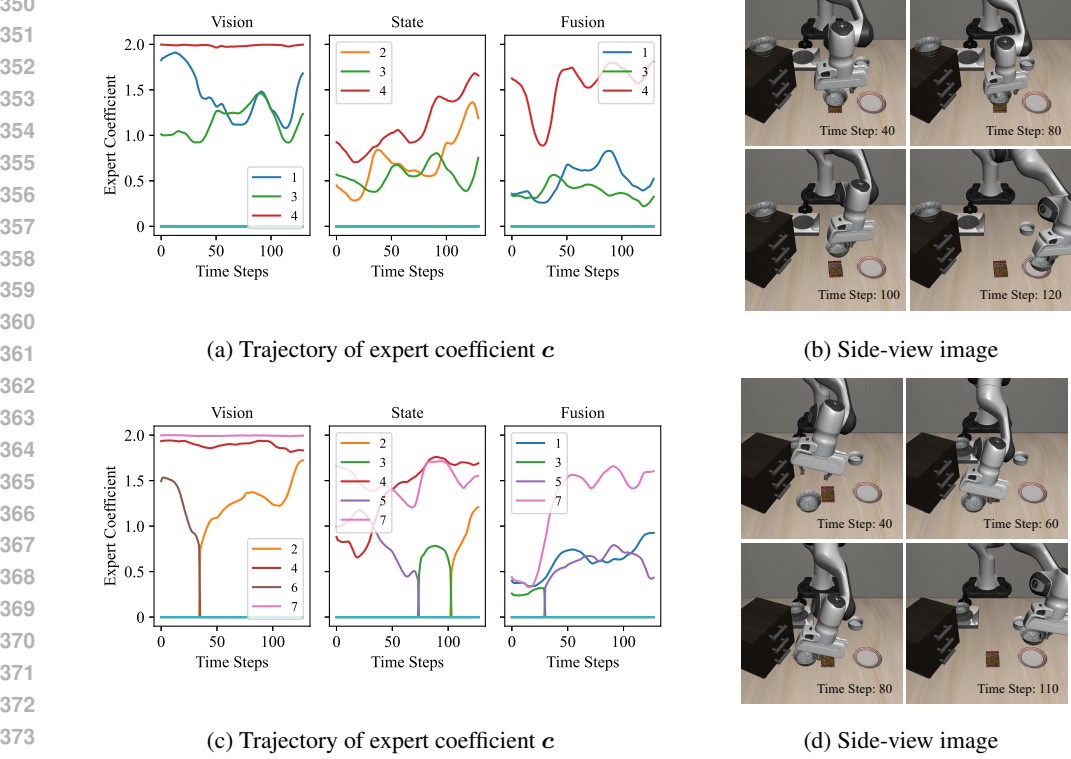

(a) Trajectory of expert coefficient $c$      (b) Side-view image

(c) Trajectory of expert coefficient $c$      (d) Side-view image

Figure 12: Visualization Analysis on Expert Activation: (a)-(b) Task 4 from LIBERO-Spatial: pick up the black bowl on the cookie box and place it on the plate; (c)-(d) Task 7 from LIBERO-Spatial: pick up the black bowl next to the cookie box and place it on the plate.

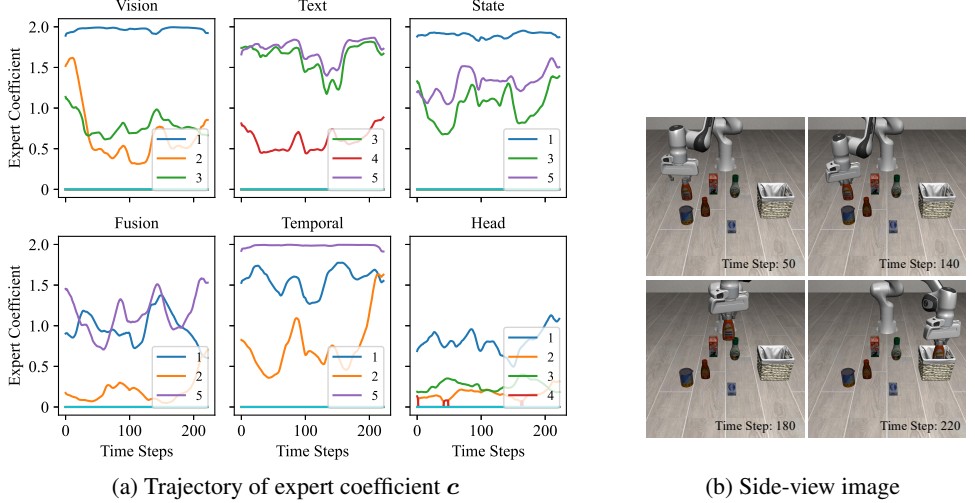

(a) Trajectory of expert coefficient $c$      (b) Side-view image

Figure 13: Visualization Analysis on Expert Activation: Task 5 from LIBERO-Object: pick up the ketchup and place it in the basket.

## B.6 LEARNING CURVES

We present the learning curves for all methods in four lifelong learning LIBERO task suites in Figure 15-18 . The $y$-axis indicates the success rate, while the $x$-axis depicts the agent's training process over the course of lifelong learning. For example, the Task 1 subplot in the figure shows the agent's performance on the first task when it learns ten tasks sequentially. Learning curves demonstrate that DMPEL achieves superior forward transfer with reduced forgetting compared to baselines.

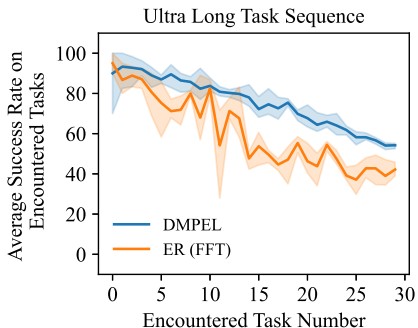

Figure 14: Average success rate on encountered tasks in the ultra long task sequence concatenated with LIBERO-Object, Goal, and Spatial benchmarks.

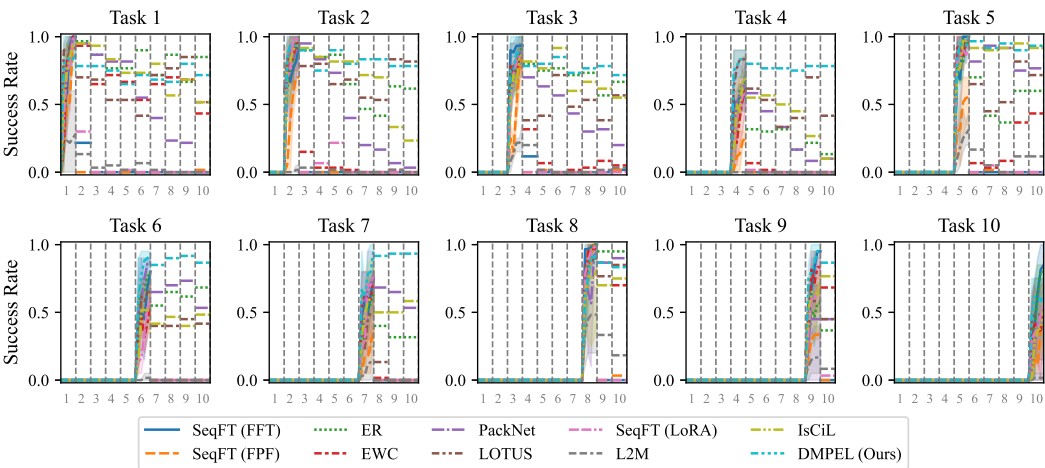

Figure 15: Visualization of learning curves on the LIBERO-Goal benchmark.

## C   DISCUSSIONS AND FUTURE DIRECTIONS

**Model Scaling and Application to Different Architectures.**   We conducted pretraining and adaptation experiments with a transformer-based model with around 200M parameters. Future work can explore scaling up to larger policies with over billions of parameters, and also applying to policies with different architectures, e.g., diffusion-based policies (Chi et al., 2025).

**Sim2Real Transfer.**   Future work can focus on addressing the following challenges for sim2real transfer: (1) Differences in Visual Input: There are substantial discrepancies between images produced by simulation engines and those captured by real-world sensors, which necessitates further fine-tuning of the visual encoder; (2) Differences Between Simulated and Real Robots: Even when utilizing the same robot model (e.g., the Franka Emika Panda), minor differences can lead to varying end-effector behaviors, underscoring the necessity of demonstrations collected on the actual robot; (3) Real-World Stochasticity: The unpredictability of real-world environments adds complexity to the transfer process. One potential solution is to incorporate a post-fine-tuning step for the pretrained policy using real-world datasets before initiating lifelong adaptation in the real environment. Given DMPEL's performance in the large-scale simulated benchmark LIBERO, we expect it to excel during the lifelong adaptation phase. However, we would like to emphasize that the sim2real domain has

Table 8: Performance on cross-domain lifelong adaptation to Robomimic ($K = 3$)

| Method | FWT | BWT | AUC |
|---|---|---|---|
| DMPEL | $0.54 \pm 0.05$ | $0.10 \pm 0.02$ | $0.54 \pm 0.02$ |

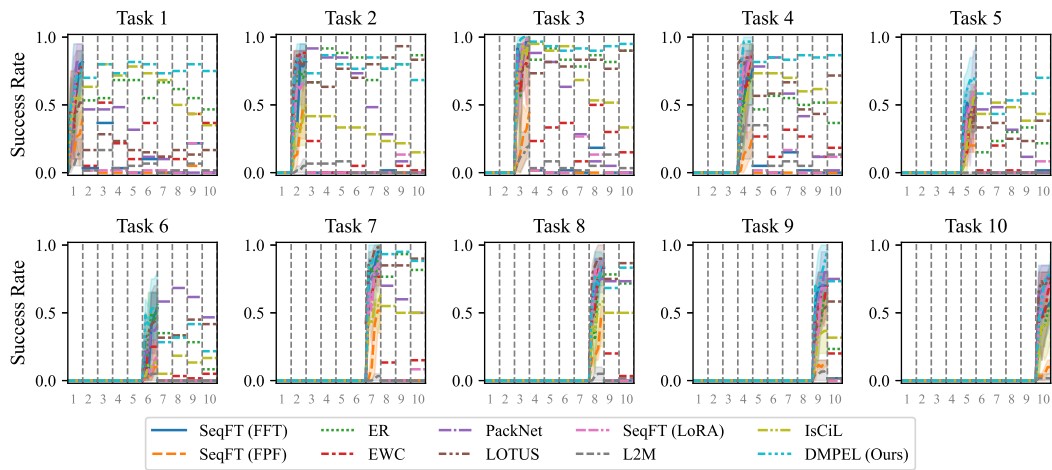

Figure 16: Visualization of learning curves on the LIBERO-Spatial benchmark.

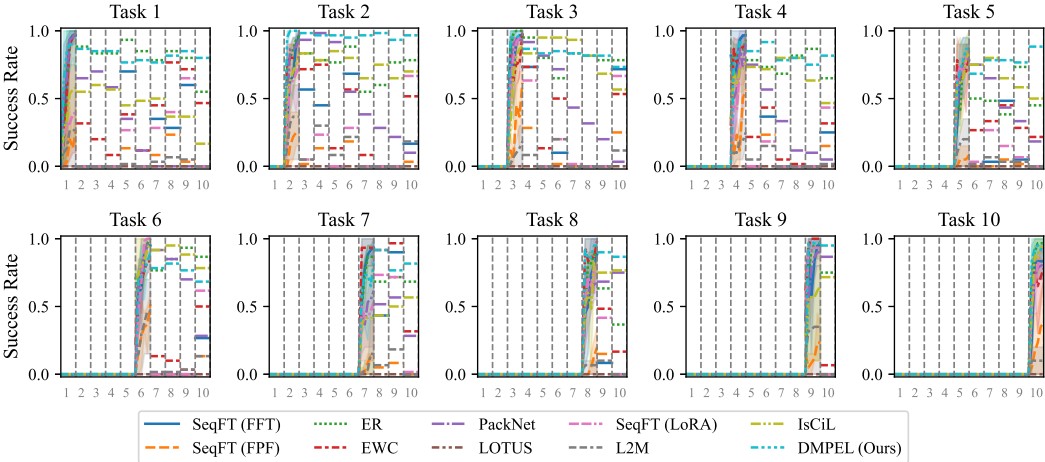

Figure 17: Visualization of learning curves on the LIBERO-Object benchmark.

a wealth of related work that is generally orthogonal to lifelong learning approaches and could be promising research directions.

**Tasks with Different Observation and Action Spaces.** In this manuscript we focus on the lifelong learning setting where the tasks share the same observation and action space. However, there are cases where these spaces may vary across different tasks, such as in cross-embodiment transfer. Techniques like Quantized Space Alignment (QSA) proposed in (Hu et al., 2025) align differing spaces through quantization methods. This approach can serve as an orthogonal technique to facilitate policy transfer across tasks with distinct observation and action spaces.

**Comparison and Integration with other Lifelong Learning Algorithms.** DMPEL is an architectural lifelong learning method that leverages LoRA modules to capture specific knowledge, while also incorporating concepts from replay and regularization methods. The expert coefficient replay technique utilizes a small number of router input-output pairs from previous tasks to regularize the router updates. Other LoRA-based baselines similarly integrate various learning algorithms, such as clustering and query-key matching for expert retrieval. Future work could explore the integration of complementary techniques to enhance lifelong learning performance, such as employing contrastive learning for representation learning and skill indexing (Choi & Seo, 2025), or using diffusion models for generative replay (Gao & Liu, 2023).

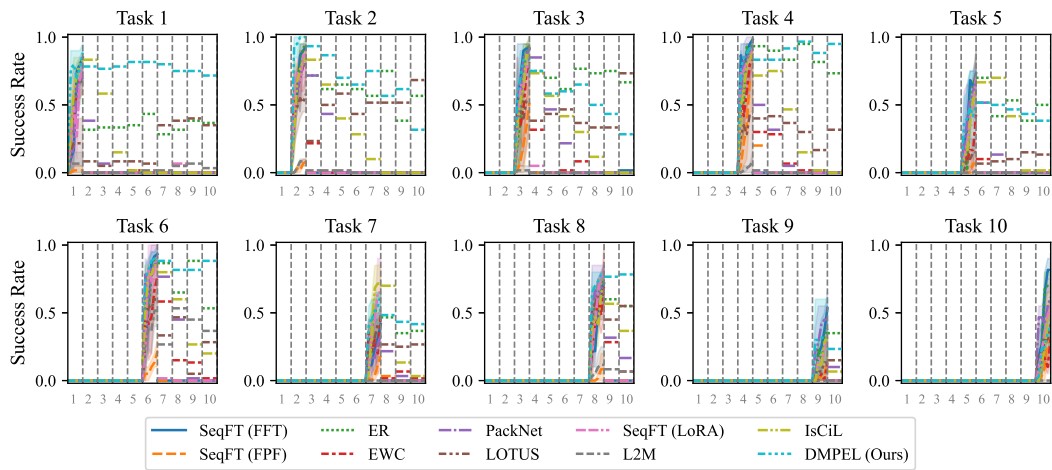

Figure 18: Visualization of learning curves on the LIBERO-Long benchmark.

**Extension beyond Imitation Learning.** In this manuscript, we adopt the lifelong imitation learning setting from LIBERO and achieve superior performance compared to other baselines. Future work could explore applying DMPEL to various robotics settings, such as lifelong reinforcement learning.

## D THE USAGE OF LARGE LANGUAGE MODELS

We only use Large Language Models (LLMs) to assist and polish writing based on prompts such as "Please help me polish the paragraph and check for any grammatical mistakes."

