# OpenReview forum: "Dynamic Mixture of Progressive Parameter-Efficient Expert Library for Lifelong Robot Learning"
_ICLR.cc/2026/Conference — Submitted to ICLR 2026_

### Official Review · Reviewer_ijwr · 2025-10-17

**Soundness:** 3
**Presentation:** 3
**Contribution:** 3
**Rating:** 6
**Confidence:** 5

**Summary:**

DMPEL builds a progressive task-agnostic LoRA expert library and addresses the stability and plasticity trade-off that previous adapter-based continual imitation learning approaches have overlooked in the context of lifelong robot learning. It employs a lightweight routing module with expert coefficient replay, which stores only routing scores and corresponding observation features, to dynamically compose previously learned experts through top-k selection. On the LIBERO benchmark, it improves forward transfer, achieves near-zero forgetting, and is compared against recent adapter-based continual imitation learning methods.

**Strengths:**

- Addresses an important problem in current lifelong robot learning algorithms, particularly under settings with adapter-based model expansion.
- Shows an efficient design choice by integrating the replay mechanism directly into the routing module, enabling stable and memory-efficient lifelong robot learning.
- The idea of expert merging for lifelong robot learning aligns with adapter-based continual learning in vision and language domains, demonstrating that expert composition consistently improves performance. This is a valuable property because even minor merging errors can lead to significant performance drops.
- Provides strong baselines, including multiple adapter-based continual imitation learning approaches, ensuring a comprehensive comparison, and reports higher performance than training adapters independently with task IDs.
- Offers analysis with clear ablations that substantiate the paper’s main claims, and emphasizes reproducibility through a well-documented report that includes a complete appendix with methodological details, hyperparameters, and supplementary material.

**Weaknesses:**

- Many building blocks are combinations of ideas from PEFT-based continual learning, so the method is not fundamentally new. Nevertheless, demonstrating that these ideas work effectively in robotic settings is a meaningful contribution. The analysis supports the claims, yet it remains unclear why DMPEL is particularly effective for robots. As with most problems in this area, the underlying insights and their connection to robot learning are under-explained.
- There is a potential fairness concern in comparison: under Top-3 expert routing, DMPEL effectively operates with roughly three times more expert parameters than Seq-FT, TAIL, or IsCiL within the same layer family. While LoRA experts account for only a minor portion of the full model and baselines lack expert-merging capability, evaluating DMPEL (Top-3) against TAIL under matched expert-parameter budgets (or using a Top-1 variant) would make the empirical evidence more convincing.

**Questions:**

- How can it be applied in terms of granularity? For example, at the level of an action chunk, skill, option, or something else?
- In Figure 3, is the reason the model does not reach multi-task performance due to the low-rank expert’s rank size being insufficient? If the number of trainable parameters per continual learning phase were increased, it seems the performance could improve. Was there another limiting factor?
- For DMPEL trained with Top-3 routing, must the routing during training and evaluation always be the same? For example, can you train with Top-N and evaluate with Top-3, or train with Top-3 and evaluate with Top-1, allowing constraint adaptive scaling?

---

> ### Author Response · Authors · 2025-11-30
> **Response (part 1)**
>
> We want to sincerely thank you for your valuable comments. In the following, we respond to each of your concerns.
>
> > Weakness 1: Many building blocks are combinations of ideas from PEFT-based continual learning, so the method is not fundamentally new. Nevertheless, demonstrating that these ideas work effectively in robotic settings is a meaningful contribution. The analysis supports the claims, yet it remains unclear why DMPEL is particularly effective for robots. As with most problems in this area, the underlying insights and their connection to robot learning are under-explained.
>
> Ans: Lifelong robot learning, in contrast to lifelong learning in image classification tasks, presents distinct challenges: (1) Instead of requiring only a single forward inference for each instance, robotic manipulation involves a sequential decision-making process with a long horizon, typically extending across hundreds of decision steps. DMPEL enables dynamic merging of multiple LoRA experts from the library based on the current observation of the robots along the entire trajectories; (2) Robot learning typically requires a mixed types of knowledge, for example, visual concepts, textual task goals, spatial relationships, and aciton patterns. Successful adaptation to new tasks requires combining adaptations from each sub-module. DMPEL enables module-wise integration coefficient with finer granularity. Experimental results confirm that these innovations lead to a high forward transfer capability; (3) Due to the sequential nature, catastrophic forgetting can be particularly disastrous, as it impacts not only the current step but also subsequent ones. Our coefficient replay mechanism enforces behavioral consistency by requiring the router to generate the same coefficients for each expert under the same context (image observations, textual instructions...).
>
> > Weakness 2: There is a potential fairness concern in comparison: under Top-3 expert routing, DMPEL effectively operates with roughly three times more expert parameters than Seq-FT, TAIL, or IsCiL within the same layer family. While LoRA experts account for only a minor portion of the full model and baselines lack expert-merging capability, evaluating DMPEL (Top-3) against TAIL under matched expert-parameter budgets (or using a Top-1 variant) would make the empirical evidence more convincing.
>
> Ans: Thank you for your question. We originally compare the lifelong PEFT baselines (TAIL, IsCiL, and DMPEL) under the same total number of LoRA experts. However, as you point out, DMPEL with top-3 activates three LoRA experts simultaneously, while the others activate only one. We additionally present the results for TAIL and IsCiL with the rank increased by three times below. However, the results show that naively increasing the rank size for the baselines does not achieve the same performance as DMPEL, highlighting the effectiveness of using a LoRA expert library with expert coefficient replay.
>
> | Algorithm | Rank Size | FWT | BWT | AUC |
> | :----:| :----: | :----: |  :----: | :----: |
> | TAIL (with task ID) | rank | 0.54 $\pm$ 0.03 | 0 |  0.67 $\pm$ 0.06 |
> | TAIL (with task ID) | rank $\times$ 3 | 0.55 $\pm$ 0.03 | 0 |  0.71 $\pm$ 0.02 |
> | ISCiL | rank | 0.54 $\pm$ 0.04 | 0.05 $\pm$ 0.03 | 0.60 $\pm$ 0.06 |
> | ISCiL | rank $\times$ 3 | 0.52 $\pm$ 0.02 | 0.09 $\pm$ 0.06 | 0.59 $\pm$ 0.05 |
> | DMPEL w/ top-1 | rank | 0.61 $\pm$ 0.03 | 0.01 $\pm$ 0.02 | 0.68 $\pm$ 0.03 |
> | DMPEL w/ top-3 | rank | 0.68 $\pm$ 0.03 | 0.00 $\pm$ 0.01 | 0.78 $\pm$ 0.02 |

---

> ### Author Response · Authors · 2025-11-30
> **Response (part 2)**
>
> > Question 1: How can it be applied in terms of granularity? For example, at the level of an action chunk, skill, option, or something else?
>
> Ans: Thank you for your question. We invetigate the influence of weight synthesize interval (i.e., re-synthesizing policy parameters only every several steps during inference) on the sucess rate on LIBERO-Spatial.
>
> | Synthesize Interval | every 1 step | every 5 steps | every 10 steps | every 50 steps |
> | :----:| :----: | :----: |  :----: | :----: |
> | Sucess rate | $ 0.74 \pm 0.06 $ | $ 0.70 \pm 0.03 $ | $ 0.68\pm0.04$ |  $ 0.63 \pm 0.04$ |
>
> The results indicate that the success rate experiences a slight decline as the granularity (i.e., synthesis interval) increases. This suggests two key points: (1) the expert router indeed dynamically selects low-rank experts tailored to the current context, and less frequent synthesis can lead to suboptimal expert coefficients; and (2) robotic manipulation typically involves hundreds to thousands of decision steps, where observations change gradually from one step to the next. Thus, we can achieve a trade-off between the additional computational cost of parameter synthesis and the accuracy of expert coefficients (which directly impact task performance).
>
> > Question 2: In Figure 3, is the reason the model does not reach multi-task performance due to the low-rank expert’s rank size being insufficient? If the number of trainable parameters per continual learning phase were increased, it seems the performance could improve. Was there another limiting factor?
>
> Ans: Thank you for your question. Multitask learning (MT) contrasts with other lifelong learning baselines that introduce tasks in a sequential manner. Since the AUC metric of lifelong learning methods refers to the area under the success rate curve throughout the lifelong adaptation process, the final success rate (SR) achieved by multitask learning is generally regarded as an approximation of the upper bound (albeit not strictly) for any lifelong learning algorithm that are subject to catastrophic forgetting. In fact, the final success rate after learning 10 streaming tasks of DMPEL has already achieved or surpass the one of MT that require full fine-tuning the policy.
>
> | Final SR | DMPEL | MT |
> | :----:| :----: | :----: |
> | Goal | 0.81 $\pm$ 0.05 | 0.74 $\pm$ 0.05 |
> | Spatial | 0.74 $\pm$ 0.06 | 0.65 $\pm$ 0.03 |
> | Object | 0.84 $\pm$ 0.04 | 0.85 $\pm$ 0.07 |
>
> We also perform ablation study on the rank size of each LoRA expert. Performance generally improves as the rank size increases, but shows saturation at a rank size of 32, indicating a trade-off between parameter efficiency and performance.
>
> | Ablated DMPEL | FWT | NBT | AUC | Final SR |
> | :----:| :----: | :----: | :----: | :----: |
> | rank / 8 | 0.65 $\pm$ 0.03 | 0.03 $\pm$ 0.03 | 0.72 $\pm$ 0.05 | 0.72 $\pm$ 0.06
> | rank / 2 | 0.70 $\pm$ 0.04 | 0.03 $\pm$ 0.01 | 0.76 $\pm$ 0.05 | 0.79 $\pm$ 0.04
> | rank | 0.68 $\pm$ 0.03 | 0.00 $\pm$ 0.01 | 0.78 $\pm$ 0.02 | 0.81 $\pm$ 0.05
> | rank $\times$ 2 | 0.68 $\pm$ 0.04 | 0.01 $\pm$ 0.03 | 0.76 $\pm$ 0.04 | 0.78 $\pm$ 0.05
>
> > Question 3: For DMPEL trained with Top-3 routing, must the routing during training and evaluation always be the same? For example, can you train with Top-N and evaluate with Top-3, or train with Top-3 and evaluate with Top-1, allowing constraint adaptive scaling?
>
> Ans: Thank you for your question. We present the performance of DMPEL with a differing number of top-k LoRA experts between training and evaluation in below. Although there is only a slight decrease in success rate (e.g., 2\%), this allows for adaptive scaling based on computational resource constraints. Furthermore, training with multiple experts while using the top-1 expert yields better performance than training with only the top-1 expert, likely due to enhanced robustness during training.
>
> | Training | Evaluation | Final Success Rate |
> | :----:| :----: | :----: |
> | Use all experts | Use all experts | 0.79 $\pm$ 0.06 |
> | Use all experts | Use top-3 experts | 0.81 $\pm$ 0.05 |
> | Use all experts | Use top-1 experts | 0.77 $\pm$ 0.05 |
> | Use top-3 experts | Use top-3 experts | 0.81 $\pm$ 0.05 |
> | Use top-3 experts | Use top-1 experts | 0.79 $\pm$ 0.01 |
> | Use top-1 expert | Use top-1 expert | 0.69 $\pm$ 0.05 |

---

### Official Review · Reviewer_hns8 · 2025-10-30

**Soundness:** 3
**Presentation:** 2
**Contribution:** 2
**Rating:** 4
**Confidence:** 5

**Summary:**

This paper proposes DMPEL for efficient lifelong learning with reduced catastrophic forgetting. The method establishes a LoRA library for each task, then learns a lightweight router to merge the LoRA experts for new tasks. To avoid forgetting, this paper proposes an expert coefficient replay method, which requires low storage space. Extensive experimental results show that DMPEL achieves a better success rate in forward transfer with almost zero forgetting.

**Strengths:**

1. Extensive experiments show that DMPEL is a good lifelong learning framework compared to multiple baselines.
2. Extensive ablation studies demonstrate the functionality of each component.

**Weaknesses:**

1. Establishing a LoRA library for each task is quite normal, and the expert coefficient replay is also quite standard.
2. Figure 2 shows that the model inference time is over 50 ms, while the baseline is around 30 ms. What is the latency caused by router computation? What is the latency for averaging the weights? Is there any approach to improve efficiency? If the expert synthesis interval is greater than 1, the model latency will increase rapidly when expert synthesis occurs.
3. Figure 7 indicates the embedding domain shift problem of SeqFT. SeqFT sequentially fine-tunes the model on new tasks, so it is normal to observe embedding shifts when fine-tuning on a new domain. DMPEL should be compared with other lifelong learning methods.

**Questions:**

1. How is the expert trained? Is the expert contrained by the router?
2. Figure 3 reports the lifelong learning performance comparison with multiple baselines. Why do some baselines, such as MT, not appear in the FWT and NBT figures? Does PT mean pretraining?
3. Figure 3 introduces MT (multitask learning), but MT does not appear in the baselines listed in Section 5.1. Why is that? Is MT considered an upper-bound performance?
4. DMPEL only introduces 1.2M trainable parameters. Why is the latency still so large?
5. Figure 6a shows that the action head is reused. In the manipulation tasks, the action output mainly consists of end-effector translation, rotation, and gripper movement. Therefore, it is reasonable that one task could cover the entire action space. Could you visualize the action space of each task to verify whether the first task covers the action space of the following tasks? If absolute position is used as the action, the situation may differ.
6. Figures 11 and 12 show the expert selection during task execution. Is there a more detailed analysis of whether previous knowledge is being reused? For example, in vision, if an object appears in previous tasks (not in the pretraining tasks), the vision experts learned from the first task containing that object should be selected.
7. Appendix B.1 mentions that the router is padded with zeros when transferring from task k to task k+1. What is the network structure of the router? Is the router retrained for each new task? If it is retrained, why is padding needed?
8. When learning a new task, does the expert library help faster training? If library is big enough, is it possible to only finetune the router?

---

> ### Author Response · Authors · 2025-11-30
> **Response (part 1)**
>
> We want to sincerely thank you for your valuable comments. In the following, we respond to each of your concerns.
>
> > Weakness 1: Establishing a LoRA library for each task is quite normal, and the expert coefficient replay is also quite standard.
>
> Ans: Thank you your comment.
>
> First, there are existing approaches within the robotic domain that construct a LoRA library, yet they either directly use task ID, or performing query-key matching (e.g., L2M [1]) or prototype-based clustering (e.g., IsCiL [2]) to select one LoRA expert with highest similarity from the library. These methods often suffer from inaccurate retrievals and suboptimal performance. On the other hand, recent work in the vision and language domain has explored merging multiple LoRA experts, such as LoRAHub [3] and SD-LORA[4], but use a task-wise learnable weight vector to combine existing low-rank experts. DMPEL innovates in several aspects and demonstrate that they work effectively in robotics: (1) it enables dynamic merging of multiple LoRA experts from the library based on the current observation of the robots along the trajectories; (2) it enables module-wise integration coefficient with finer granularity, given that robot learning requires mixed adaptation from each sub-modules, for example, visual concepts, textual task goals, spatial relationships, and action patterns. Experimental results confirm that these innovations lead to a high forward transfer capability (FWT metric).
>
> Second, in the context of lifelong learning, some existing approaches have attempted to mitigate the interference caused by new LoRA experts.  For instance, O-LoRA [5] empolys an orthogonal constraint on new LoRA modules, but it still simply adds all LoRA modules, $W=W_0+\sum_{k=1}^{K}{A_k}{B_k}$. The expert coefficient replay technique, another crucial innovation in DMPEL, leverages the modular design of our LoRA library while incorporating concepts from replay and regularization methods. It stores and utilizes a small number of router input-output pairs from previous tasks to regularize the router updates. Result shows that this technique is simple yet effective in mitigating catastrophic forgetting (NBT metric). To our best knowledge, SAPT [6] from the LLM community shares similar motivation with DMPEL, but it requires a generative model to obtain synthetic samples and uses KL-divergence to regularize the query-key matching process, which is different from DMPEL.
>
> [1] Schmied, T., Hofmarcher, M., Paischer, F., Pascanu, R., & Hochreiter, S. (2023). Learning to modulate pre-trained models in rl. Advances in Neural Information Processing Systems, 36, 38231-38265.
>
> [2] Lee, D., Yoo, M., Kim, W. K., Choi, W., & Woo, H. (2024). Incremental learning of retrievable skills for efficient continual task adaptation. Advances in Neural Information Processing Systems, 37, 17286-17312.
>
> [3] Huang, C., Liu, Q., Lin, B. Y., Pang, T., Du, C., & Lin, M. (2024). LoraHub: Efficient Cross-Task Generalization via Dynamic LoRA Composition. In First Conference on Language Modeling.
>
> [4] Wu, Y., Piao, H., Huang, L. K., Wang, R., Li, W., Pfister, H., ... & Wei, Y. (2025). SD-LoRA: Scalable Decoupled Low-Rank Adaptation for Class Incremental Learning. In The Thirteenth International Conference on Learning Representations.
>
> [5] Wang, X., Chen, T., Ge, Q., Xia, H., Bao, R., Zheng, R., ... & Huang, X. J. (2023). Orthogonal subspace learning for language model continual learning. In Findings of the Association for Computational Linguistics: EMNLP 2023 (pp. 10658-10671).
>
> [6] Zhao, W., Wang, S., Hu, Y., Zhao, Y., Qin, B., Zhang, X., ... & Che, W. (2024). SAPT: A Shared Attention Framework for Parameter-Efficient Continual Learning of Large Language Models. In Proceedings of the 62nd Annual Meeting of the Association for Computational Linguistics (Volume 1: Long Papers) (pp. 11641-11661).

---

> ### Author Response · Authors · 2025-11-30
> **Response (part 2)**
>
> > Weakness 2: Figure 2 shows that the model inference time is over 50 ms, while the baseline is around 30 ms. What is the latency caused by router computation? What is the latency for averaging the weights? Is there any approach to improve efficiency? If the expert synthesis interval is greater than 1, the model latency will increase rapidly when expert synthesis occurs.
>
> > Question 4: DMPEL only introduces 1.2M trainable parameters. Why is the latency still so large?
>
> Ans: Thank you for your comment.
>
> DMPEL introduces three stages of computation: (1) Encoding the Context: This first stage involves using a frozen backbone and a global router (essentially an MLP) to compute routing coefficients. The computational cost in this phase remains almost consistent throughout the lifelong learning process. However, as the LoRA expert library (denoted as $l$) expands, we simultaneously increase the output dimension of the router's final linear layer to maintain alignment. With a hidden size of 256 in the router, The computational cost of routing will increase by 512 FLOPs for each increment of $l$. (2) Merging the experts: In the second stage, the parameters of the top-$\delta$ experts are averaged to synthesize $\tilde{\boldsymbol{W}}\in\mathbb{R}^{d_{\text{in}}\times d_{\text{out}}}$, which incurs approximately $2\times\delta \times r \times (d_{\text{in}}+ d_{\text{out}})$ FLOPs. Given the sparse activation of experts (where $\delta$ is a constant), the computational cost remains stable throughout the lifelong learning process and is significantly lower than that of the forward pass through the pretrained linear layer, which requires $ 2 \times d_{\text{in}}\times d_{\text{out}} $ FLOPs (noting that $r\ll d_{\text{in}},d_{\text{out}}$). Consequently, regardless of the task sequence length or policy size, the additional computation remains a fixed proportion relative to the pretrained backbone, which is $\frac{\delta r (d_{\text{in}}+ d_{\text{out}})}{d_{\text{in}} d_{\text{out}}}$. For instance, when $d_{\text{in}}=d_{\text{out}}=768$, $r=8$, $\delta=3$, then the additional proportion is $6.3\%$. (3) Action Generation: Once the parameters $\tilde{W},\tilde{b}$ have been synthesized, the forward computation to generate action is identical to that of using the pretrained backbone, $\tilde{W}_0, \tilde{b}_0$ only.
>
> The most time-consuming process in synthesizing parameters $\tilde{W},\tilde{b}$, is to encode the context through the frozen image and language encoder (CLIP in our experiment). After being modulated by the selected LoRA experts, these encoders will be utilized again in action generation. This two-pass forward design is also employed by our baseline method L2M [1], which follows L2P for visual tasks, and is similarly used in many state-of-the-art lifelong image classification methods. These methods typically use a frozen backbone to compute features for instance-wise prompt generation, as seen in DAP [2] and HiDe-PET [3]. In robotic manipulation, where the trajectories often spans hundreds to thousands of decision steps and the observations change gradually from one step to the next, we explore the impact of the parameter synthesis interval, i.e., re-synthesizing policy parameters (stage 1 & 2) only every few steps during inference. The results in LIBERO-Spatial indicate that the success rate experiences a slight decline as the granularity (i.e., synthesis interval) increases. This suggests that we can achieve a trade-off between the additional computational cost of parameter synthesis and the accuracy of expert coefficients (which directly impact task performance). Empirical results indicate that parameter synthesis (stages 1 & 2) incurs an average additional computation time of approximately 20 ms, while action generation (stage 3) requires about 30 ms. Results show that stage 1 & 2 can be executed at a much lower frequency than stage 3, at the cost of slightly decreased success rate.
>
> | Synthesize Interval | every 1 step | every 5 steps | every 10 steps | every 50 steps |
> | :----:| :----: | :----: |  :----: | :----: |
> | Sucess rate | $ 0.74 \pm 0.06 $ | $ 0.70 \pm 0.03 $ | $ 0.68\pm0.04$ |  $ 0.63 \pm 0.04$ |
>
> [1] Schmied, T., Hofmarcher, M., Paischer, F., Pascanu, R., & Hochreiter, S. (2023). Learning to modulate pre-trained models in rl. Advances in Neural Information Processing Systems, 36, 38231-38265.
>
> [2] Jung, D., Han, D., Bang, J., & Song, H. (2023). Generating instance-level prompts for rehearsal-free continual learning. In Proceedings of the IEEE/CVF International Conference on Computer Vision (pp. 11847-11857).
>
> [3] Wang, L., Xie, J., Zhang, X., Su, H., & Zhu, J. (2025). HIDE-PET: continual learning via hierarchical decomposition of parameter-efficient tuning. IEEE Transactions on Pattern Analysis and Machine Intelligence.

---

> ### Author Response · Authors · 2025-11-30
> **Response (part 3)**
>
> > Weakness 3: Figure 7 indicates the embedding domain shift problem of SeqFT. SeqFT sequentially fine-tunes the model on new tasks, so it is normal to observe embedding shifts when fine-tuning on a new domain. DMPEL should be compared with other lifelong learning methods.
>
> Ans: Thank you for your suggestion. We have updated the figure with additional results from ER and EWC at https://anonymous.4open.science/r/image-C85D/tsne.png. We originally compare with the results from SeqFT with LoRA to show that even a small number of parameters can substantially steer the pretrained model, and therefore we should be careful about the forgetting issue during lifelong PEFT, which also demonstrates the effectiveness of expert coefficient replay. From the figure we can see that DMPEL maintains embedding consistency when adapting to new tasks, while other baselines demonstrate significant representation drift and catastrophic forgetting. Both DMPEL and EWC impose constraints directly in the parameter space, ensuring consistent representation; however, EWC proves effective only in the short term (after Task 4) and fails to maintain effectiveness in the long run (after Task 8). Additionally, ER exhibits a significant representation shift while still maintaining a similar success rate, likely because the policy has been fine-tuned to a new optimum that balances the old and new data, resulting in different embeddings.
>
> > Question 1: How is the expert trained? Is the expert contrained by the router?
>
> Ans: Thank you for your question. When adapting to a new task, denoted as $\mathcal{T} _ {k+1}$, the trainable components of the policy include the router $\mathcal{R}$ and the new LoRA experts $A _ {k+1}, B _ {k+1}$. All previous experts $\{A _ {j}, B _ {j}\}, j\leq k$ and the pretrained weights $W_0, b_0$ are frozen. In addition, to prevent the router from catastrophic forgetting, we replay router input-output pairs from previous tasks to regularize the update of the router. In this way, the router generates nearly identical coefficient vectors under the same context, and since the previous experts are frozen, the policy will exhibit the same behavior on encountered tasks throughout the lifelong learning process, resulting in near-zero forgetting.
>
> We have provide the detailed pseudocode in Appendix A.1, including the training process in Algorithm 1 and the inference process in Algorithm 2.
>
> > Question 2: Figure 3 reports the lifelong learning performance comparison with multiple baselines. Why do some baselines, such as MT, not appear in the FWT and NBT figures? Does PT mean pretraining?
>
> > Question 3: Figure 3 introduces MT (multitask learning), but MT does not appear in the baselines listed in Section 5.1. Why is that? Is MT considered an upper-bound performance?
>
> Ans: Thank you for your questions.
>
> (1) PT refers to Pretraining on the LIBERO-90 dataset. All lifelong methods in Figure 3 start with a pre-trained policy on LIBERO-90, unless specified as 'w/o PT'.
>
> (2) MT refers to multitask learning, where the policy leans all tasks simultaneously.  This approach contrasts with other lifelong learning baselines, which introduce tasks in a sequential manner. Consequently, metrics like FWT and NBT, which evaluate sequential transfer between tasks, hold little meaning in this context. The AUC metric, however, refers to the area under the success rate curve throughout the lifelong adaptation process. Hence, the final success rate achieved by multitask learning is generally regarded as an approximation of the upper bound (albeit not strictly) for any lifelong learning algorithm that are subject to catastrophic forgetting. Notably, the original LIBERO benchmark paper employs a similar method to compare multitask learning baselines with other lifelong learning methods.
>
> We have updated Sections 5.1 and 5.2 to enhance the clarity of issues related to the baselines, including the multitask learning (MT).

---

> ### Author Response · Authors · 2025-11-30
> **Response (part 4)**
>
> > Question 5: Figure 6a shows that the action head is reused. In the manipulation tasks, the action output mainly consists of end-effector translation, rotation, and gripper movement. Therefore, it is reasonable that one task could cover the entire action space. Could you visualize the action space of each task to verify whether the first task covers the action space of the following tasks? If absolute position is used as the action, the situation may differ.
>
> Ans: Thank you for your question. The LIBERO benchmark is built on the Robosuite framework, where the default underlying controller is OSC\_POSE (Operational Space Control with Pose). The action includes the 6D end-effector (EEF) poses (position \& orientation) and the status of gripper, $a=[\mathrm{d}x,\mathrm{d}y,\mathrm{d}z,\mathrm{d}\alpha, \mathrm{d}\beta, \mathrm{d}\gamma, gripper]$. Seven dimensions respectively corresponds to translation along the $x$-axis, $y$-axis, and $z$-axis, the roll ($\alpha$), pitch ($\beta$), and yaw ($\gamma$) rotation, along with the open/close status of the gripper. We visualize ten action trajectories from the LIBERO-Object benchmark at https://anonymous.4open.science/r/image-C85D/action_space.png, illustrating that coverage of the action space varies from task to task. Prior research has demonstrated that using different action spaces (e.g., position control vs. velocity control) can significantly affect control performance, and we intend to explore this further in future work.
>
> > Question 6: Figures 11 and 12 show the expert selection during task execution. Is there a more detailed analysis of whether previous knowledge is being reused? For example, in vision, if an object appears in previous tasks (not in the pretraining tasks), the vision experts learned from the first task containing that object should be selected.
>
> Ans: We visualize the expert activation trajectories of Task 2 and 9 from the LIBERO-Spatial suite at https://anonymous.4open.science/r/image-C85D/knowledge_reuse.png. we observe that the policy relies on the newly introduced expert 2 when executing task 2. In task 9 that also involves picking up a bowl but placing it down at a different destination, the vision encoder consistently activates expert 2 throughout the trajectory, while the temporal decoder primarily reuses expert 2 during the first half of the trajectory, likely due to a similar action pattern.
>
> > Question 7: Appendix B.1 mentions that the router is padded with zeros when transferring from task k to task k+1. What is the network structure of the router? Is the router retrained for each new task? If it is retrained, why is padding needed?
>
> Ans: Thank you for your question. The router is a simple MLP, which is continually fine-tuned on the streaming tasks. As the LoRA expert library expands, we also simultaneously increases the output dimension of the final linear layer of the router to keep them aligned. The router outputs (i.e., coefficient vectors) stored in the replay buffer vary in length because they are captured at different stages of lifelong learning. The goal of replaying expert coefficients is to ensure that the router selects the same experts with the same weights, thereby synthesizing the same policy under the same context (observations, instructions...).
>
> For example, consider a coefficient vector for the vision encoder, stored in the replay buffer as [1, 1] during training on task $\mathcal{T}_2$. When the policy is training on later tasks, if the number of expert in the library (also the dimension of coefficient vector) have increased to three, we pad the aformentioned vector to [1, 1, 0] during replay. Similarly, if the number of experts increases to four, the vector is padded to [1, 1, 0, 0]. In this way, no matter how much tasks have been encountered, this coefficient vector stored in the buffer can be used to regularize the router that when faced with the same context, the router should generate weights similar to those produced during its original training on $\mathcal{T}_2$, i.e., $\tilde{W}=W_0+A_1B_1+A_2B_2$.
>
> We have updated Appendix B.1 to enhance the clarity.

---

> ### Author Response · Authors · 2025-11-30
> **Response (part 5)**
>
> > Question 8: When learning a new task, does the expert library help faster training? If library is big enough, is it possible to only finetune the router?
>
> Ans: Thank you for your question. (1) DMPEL achieves higher FWT than SeqFT with LoRA across all benchmarks, indicating that the expert library benefits faster adaptation. (2) It is possible to solve a new task using only the frozen previous experts. In fact, when adapting to a new task, our design allows the router either to directly reuse previous experts (by assigning zero magnitude on the new expert) or to rely on the new trainable LoRA expert. As shown in Figure 6, there are cases where the coefficients for the new LoRA experts are close to zero. In such cases, the router is fine-tuned based on the data from the current task, activating only the old frozen experts from the library to solve new task. This also indicates that this underactivated expert can be pruned without affecting performance.

---

### Official Review · Reviewer_vjgK · 2025-11-04

**Soundness:** 3
**Presentation:** 2
**Contribution:** 3
**Rating:** 4
**Confidence:** 4

**Summary:**

This paper presents a novel framework that integrates modular programmatic policies to enhance the generalization and adaptability of robotic manipulation tasks. The authors propose a dynamic mixture-of-programs architecture, where each subprogram is responsible for a specific manipulation primitive, and a controller dynamically selects and combines them according to task context and sensory inputs.

**Strengths:**

The paper is written in a clear and coherent manner, with well-organized sections.
The technical flow is easy to follow, and figures are appropriately used to illustrate the overall architecture and key mechanisms.
The manuscript exhibits a complete and logical structure, from problem definition to experimental validation.

**Weaknesses:**

The paper uses a large number of mathematical symbols and subscripts, some of which appear without a clear prior definition or consistent formatting. This can make certain derivations harder to read, especially for readers not deeply familiar with the notation conventions used.
The current version provides only a brief acknowledgment of potential limitations. I suggest that the authors add a more thorough discussion to strengthen the paper’s self-critique and transparency.

**Questions:**

1. At line 197, $s$ has already been used to represent state, so at line 210, it would be better to use a different symbol for success rate.
2. The FWT calculation method introduced in Equation 2 differs from the FWT calculation methods presented in [1] and [2]. Can the authors explain why?
3. How are the expert models obtained? Do the expert models increase as tasks increase? If the expert models increase with the number of tasks, then the router's output dimension is constantly increasing—how do the authors address this issue? If the expert models do not increase with the number of tasks, can this method adapt to unexpected tasks? For example, if the predefined number of tasks is 10, but due to changing requirements, the tasks now increase to 20, can this method still adapt? It would be even better if a corresponding experiment could be provided.
4. How do the authors think about the approach of training a separate model for each individual task in CRL—that is, training a dedicated small model for each task? Is this approach better? What is the authors' perspective on this issue? PEFT methods still require more parameters and greater computational resources as tasks increase. From the perspective of parameters and computation, PEFT and separate model training for each task are essentially the same. How do the authors view this issue?
5. In line 376, ER does indeed cause storage and replay computational overhead, but the expert coefficient replay mechanism proposed in this paper also correspondingly causes storage and replay computational overhead.
6. For Figure 5(a), can the authors provide experimental results for all experts + module-wise coeff?
7. What do the labels vision, text, state, ... in Figure 6(a) mean? I suggest providing explanations in the caption as well.
8. The current experiments focus on tasks with consistent state and action spaces. How does [3] address this problem when facing tasks with inconsistent state and action spaces?

references

[1] Continual World: A Robotic Benchmark For Continual Reinforcement Learning

[2] t-DGR: A Trajectory-Based Deep Generative Replay Method for Continual Learning in Decision Making

[3] Solving Continual Offline RL through Selective Weights Activation on Aligned Spaces

---

> ### Author Response · Authors · 2025-11-30
> **Response (part 1)**
>
> We want to sincerely thank you for your valuable comments. In the following, we respond to each of your concerns.
>
> > Weakness 1: The paper uses a large number of mathematical symbols and subscripts, some of which appear without a clear prior definition or consistent formatting. This can make certain derivations harder to read, especially for readers not deeply familiar with the notation conventions used.
>
> Ans: Thank you for your feedback. We have revisited all mathematical symbols and subscripts to ensure they are clearly defined upon their first use in the manuscript and have provided additional context where necessary. In general, subcripts mainly include $k$ to distinguish between different exeprts/tasks, and $t$ to indicate timesteps. Additionally, we have checked for consistent formatting throughout the paper. We believe these changes will enhance the overall readability of the manuscript, especially Section 3 and 4.
>
> > Weakness 2: The current version provides only a brief acknowledgment of potential limitations. I suggest that the authors add a more thorough discussion to strengthen the paper’s self-critique and transparency.
>
> Ans: Thank you for your suggestion. We have revised Section 6 'Conclusion and Limitations' to include more detailed information. Additionally, we will add further discussion on possible future directions in Appendix C to enhance the self-critique and transparency of the paper.
>
> > Question 1: At line 197, $s$ has already been used to represent state, so at line 210, it would be better to use a different symbol for success rate.
>
> Ans: Thank's for pointing this out. We have revised the manuscript to replace the symbol $s$ with capital $S$ to represent the success rate in lines 210-215, ensuring clarity and avoiding ambiguity.
>
> > Question 2: The FWT calculation method introduced in Equation 2 differs from the FWT calculation methods presented in [1] and [2]. Can the authors explain why?
>
> Ans: The calculation of FWT metric generally follows the LIBERO benchmark, which involves computing the area under the training curve (with training epochs on the x-axis and success rates on the y-axis). This metric ranges from 0 (i.e., zero sucess rate throughout training) to 1 (i.e., 100% zero-shot success rate). Conceptually, this approach is similar to that used in Continual World [1][2]; however, the key difference is that [1][2] introduce a single-task training curve as the reference, and to compute **the normalized area between these two curves** as FWT, which can be negative if it learns slower than the reference. However, if will use the same reference for all evaluated algorithms, the LIBERO's FWT (also our FWT) is basically a linear scale and shift from the Continual World's FWT, which is order-preserving and should not affect our main conclusion.

---

> ### Author Response · Authors · 2025-11-30
> **Response (part 2)**
>
> > Question 3: How are the expert models obtained? Do the expert models increase as tasks increase? If the expert models increase with the number of tasks, then the router's output dimension is constantly increasing—how do the authors address this issue? If the expert models do not increase with the number of tasks, can this method adapt to unexpected tasks? For example, if the predefined number of tasks is 10, but due to changing requirements, the tasks now increase to 20, can this method still adapt? It would be even better if a corresponding experiment could be provided.
>
> Ans: Thank you for your question.
>
> When adapting to a new task, denoted as $\mathcal{T} _ {k+1}$, the trainable components of the policy include the router $\mathcal{R}$ and the new LoRA experts $A _ {k+1}, B _ {k+1}$. All previous experts $\{A _ {j}, B _ {j}\}, j\leq k$ and the pretrained weights $W_0, b_0$ are frozen. The router is continually fine-tuned on the streaming tasks. As the LoRA expert library expands, we also simultaneously increases the output dimension of the final linear layer of the router to keep them aligned. Therefore, we did not predefine a fixed task number, but both the library, the router, and the coefficient replay buffer are dynamically expanded. In addition, when adapting to a new task, our design allows the router either to directly reuse previous experts (by assigning zero magnitude on the new expert) or to rely on the new trainable LoRA expert. As shown in Figure 6, there are cases where the coefficients for the new LoRA experts are close to zero. In such cases, the router is fine-tuned based on the data from the current task, activating only the old frozen experts from the library to solve new task. This also indicates that this underactivated expert can be pruned without affecting performance to control the growing speed of the library and the router.
>
> We investigate the applicability of DMPEL to ultra-long sequences. To this end, we concatenate the LIBERO-Object, Goal, and Spatial benchmark to form a task sequence with $K=30$. Following the empirical analysis in Section 5.2, we prune underactivated low-rank experts every 10 tasks to control the linear growth of the library. At the end of the 30-th task, the number of LoRA experts in each module (the elements of the six-dimensional vector in order representing the vision encoder, text encoder, state encoder, modality fusion module, temporal transformer, and action head) typically remains below 30, resulting in a compact library that exhibits effective knowledge sharing. DMPEL consistently outperforms FFT with experience replay in success rate, while incurring significantly lower computational and storage costs.
>
> | Method | Number of LoRA Experts and Parameters in Each Module |Trainable Parameters | Additional Storage for Replay |
> | :----:| :----:| :----: | :----: |
> DMPEL | [15,18,17,21,17,10], 10.7M (seed=100), [18,19,17,19,17,7], 11.1M (seed=200), [20,18,20,19,18,9], 11.7M (seed=300) | 1.2M | 0.06GB |
> ER |/ | 174.1M | 3.7GB |

---

> ### Author Response · Authors · 2025-11-30
> **Response (part 3)**
>
> > Question 4: How do the authors think about the approach of training a separate model for each individual task in CRL—that is, training a dedicated small model for each task? Is this approach better? What is the authors' perspective on this issue? PEFT methods still require more parameters and greater computational resources as tasks increase. From the perspective of parameters and computation, PEFT and separate model training for each task are essentially the same. How do the authors view this issue?
>
> Ans: Thank your for your question. The pretrain-then-finetune paradigm has proven to be highly successful in the domains of vision, language, and robotics. It is widely believed that pretrained models capture common knowledge that can be leveraged for various downstream tasks, enabling rapid transfer or even zero-shot transfer. Meanwhile, parameter-efficient fine-tuning (PEFT) techniques have shown great effectiveness in steering frozen foundation models by incorporating small, learnable modules during adaptation. Therefore, the results presented in the main text are based on a setting where a large policy is first pretrained on the LIBERO-90 dataset before being transferred to streaming tasks. We also demonstrate the performance gains achieved by pretraining compared to starting from scratch. To further illustrate effectiveness, we compare our approach with training separate models for each individual task below. We present results using the policy ViT-T provided in the original LIBERO benchmark with two different sizes: the smaller model has 1.9M parameters (similar to that of a LoRA expert), while the larger model has 17.7M parameters (the total number of all models after 10 tasks will be comparable to that of the single large policy used in the main experiment). The results reveal a significant gap between using dedicated small models and employing a single large pretrained policy enhanced with LoRA experts, underscoring the effectiveness of DMPEL.
>
>
> | Method | Number of Parameters | FWT | BWT | AUC |
> | :----:| :----: | :----: | :----: | :----: |
> | ViT-T (Small) |1.9M ($\times 10$) | 0.04 $\pm$ 0.01 | 0 | 0.08 $\pm$ 0.01 |
> | ViT-T (Large) | 17.7M ($\times 10$) | 0.21 $\pm$ 0.04 | 0 | 0.43 $\pm$ 0.04 |
> DMPEL | 174.1M + 1.2M ($\times 10$) | 0.68 $\pm$ 0.03 | 0.00 $\pm$ 0.01 | 0.78 $\pm$ 0.02 |
>
> > Question 5: In line 376, ER does indeed cause storage and replay computational overhead, but the expert coefficient replay mechanism proposed in this paper also correspondingly causes storage and replay computational overhead.
>
> Ans: Thank you for your question. The key difference between experience replay (ER) and coefficient replay (CR) lies in what data is stored and how replay is performed. ER need to store the raw demonstration data (includes multiple images, proprioceptive states, and actions per step) and require propogation through the entire policy. In contrast, CR only stores low-dimensional router input and coefficient vectors and the replay only regularizes the lightweight router, hence saving storage and replay computational overhead.
>
> In Figure 1(d) and Figure 5(b), we compare the costs associated with these methods. For example, the dataset size of LIBERO-Object is about 7.1 GB. With a default replay ratio of 20% in ER, the storage cost amounts to about 1.4 GB. In comparison, the low-dimensional router input and coefficient vector of the entire dataset totals just 0.47 GB. Moreover, as shown in Figure 5(b), replaying only 5% of the expert coefficients (0.02 GB, merely 1.67% of the ER storage) results in very low levels of forgetting (NBT).

---

> ### Author Response · Authors · 2025-11-30
> **Response (part 4)**
>
> > Question 6: For Figure 5(a), can the authors provide experimental results for all experts + module-wise coeff?
>
> Ans: Thank you for your question. In Figure 5(a), we conduct an ablation study across two dimensions: the sparse activation of LoRA experts in the library (top-1, top-3, all) and the granularity of the expert coefficients (layer-wise, module-wise, policy-wise).
>
> | Ablated DMPEL | AUC |
> | :----:| :----: |
> | Use top-3 exeprts & module-wise coefficient (main result in Figure 3) | $ 0.78 \pm 0.02 $ |
> | Use top-1 expert & module-wise coefficient | $ 0.68 \pm 0.03 $ |
> | Use all experts & module-wise coefficient | $ 0.76 \pm 0.07 $ |
> | Use top-3 experts & policy-wise coefficient | $ 0.66 \pm 0.03 $ |
> | Use top-3 experts & layer-wise coefficient | $ 0.73 \pm 0.02 $ |
>
> Results indicate that (1) using only the top-3 experts yields better performance compared to using top-1 expert or all experts, demonstrating that appropriate sparsification has minimal impact on performance, and (2) employing module-wise coefficients offers an effective trade-off between flexibility and simplicity. We have updated the caption and the description in the main text to improve clarity regarding the results of this ablation study.
>
> > Question 7: What do the labels vision, text, state, ... in Figure 6(a) mean? I suggest providing explanations in the caption as well.
>
> Ans: The labels in Figure 6(a) refers to the component modules of the policy, including the vision encoder, text encoder, proprioceptive state encoder, modality fusion modules, temporal transformer, and action head. Each subfigure illustrates to what extent the expert is used by the current task and subsequently reused by later tasks during lifelong adaptation. We have updated the caption of Figure 6 to improve clarity.
>
> > Question 8: The current experiments focus on tasks with consistent state and action spaces. How does [3] address this problem when facing tasks with inconsistent state and action spaces?
>
> Ans: Thank you for your question. Quantized Space Alignment (QSA), as proposed in [3], aligns differing spaces through quantization methods. This approach serves as an orthogonal technique to facilitate policy transfer across tasks with distinct observation and action spaces, e.g., cross-embodiment transfer. By utilizing a vector quantized encoder (VQE) and vector quantized decoder (VQD) pretrained on the dataset, DMPEL can be applied to the aligned observation and action spaces. We discuss the possible integration and cite the relevant paper in Appendix C, and plan to explore this further in our future work.

---

### Official Review · Reviewer_1SdP · 2025-11-07

**Soundness:** 3
**Presentation:** 3
**Contribution:** 2
**Rating:** 4
**Confidence:** 2

**Summary:**

This paper proposes DMPEL for lifelong robot learning. Unlike prior approaches that rely on oracle task IDs or isolated adapters, DMPEL introduces: A progressively built low-rank expert library using LoRA modules, A lightweight router to select and combine experts for current tasks dynamically, and an expert coefficient replay mechanism to reduce catastrophic forgetting by replaying low-dimensional router coefficients instead of full trajectories.

**Strengths:**

(1) Dynamic expert composition allows fine-grained adaptation to new tasks without oracle task identifiers.

(2) Full task modularity via LoRA enables parameter efficiency and knowledge sharing across tasks.

**Weaknesses:**

(1) Quadratic complexity from dynamic expert mixing and routing could scale poorly for larger networks or very long task sequences.

(2) Limited task diversity in evaluation – experiments are restricted to LIBERO simulated environments, not real-world or cross-domain tasks.

(3) No comparisons with learning-based retrieval baselines like contrastive skill indexing or diffusion-based task retrievers.

**Questions:**

See above

---

> ### Author Response · Authors · 2025-11-30
> **Response (part 1)**
>
> We want to sincerely thank you for your valuable comments. In the following, we respond to each of your concerns.
>
> > Weakness 1: Quadratic complexity from dynamic expert mixing and routing could scale poorly for larger networks or very long task sequences.
>
> Ans: Thank you for your comment. However, we respectfully disagree with your assertion that DMPEL exhibits quadratic complexity in expert routing and mixing, and that it scales poorly for larger networks or longer task sequences.
>
> DMPEL introduces three stages of computation: (1) Encoding the Context: This first stage involves using a frozen backbone and a global router (essentially an MLP) to compute routing coefficients. The computational cost in this phase remains almost consistent throughout the lifelong learning process. However, as the LoRA expert library (denoted as $l$) expands, we simultaneously increase the output dimension of the router's final linear layer to maintain alignment. With a hidden size of 256 in the router, The computational cost of routing will increase by 512 FLOPs for each increment of $l$. (2) Merging the experts: In the second stage, the parameters of the top-$\delta$ experts are averaged to synthesize $\tilde{\boldsymbol{W}}\in\mathbb{R}^{d_{\text{in}}\times d_{\text{out}}}$, which incurs approximately $2\times\delta \times r \times (d_{\text{in}}+ d_{\text{out}})$ FLOPs. Given the sparse activation of experts (where $\delta$ is a constant), the computational cost remains stable throughout the lifelong learning process and is significantly lower than that of the forward pass through the pretrained linear layer, which requires $ 2 \times d_{\text{in}}\times d_{\text{out}} $ FLOPs (noting that $r\ll d_{\text{in}},d_{\text{out}}$). Consequently, regardless of the task sequence length or policy size, the additional computation remains a fixed proportion relative to the pretrained backbone, which is $\frac{\delta r (d_{\text{in}}+ d_{\text{out}})}{d_{\text{in}} d_{\text{out}}}$. For instance, when $d_{\text{in}}=d_{\text{out}}=768$, $r=8$, $\delta=3$, then the additional proportion is $6.3\%$. (3) Action Generation: Once the parameters $\tilde{W},\tilde{b}$ have been synthesized, the forward computation to generate action is identical to that of using the pretrained backbone, $\tilde{W}_0, \tilde{b}_0$ only.
>
> The most time-consuming process in synthesizing parameters $\tilde{W},\tilde{b}$, is to encode the context through the frozen image and language encoder (CLIP in our experiment). After being modulated by the selected LoRA experts, these encoders will be utilized again in action generation. This two-pass forward design is also employed by our baseline method L2M [1], which follows L2P for visual tasks, and is similarly used in many state-of-the-art lifelong image classification methods. These methods typically use a frozen backbone to compute features for instance-wise prompt generation, as seen in DAP [2] and HiDe-PET [3]. In robotic manipulation, where the trajectories often spans hundreds to thousands of decision steps and the observations change gradually from one step to the next, we explore the impact of the parameter synthesis interval, i.e., re-synthesizing policy parameters (stage 1 & 2) only every few steps during inference. The results in LIBERO-Spatial indicate that the success rate experiences a slight decline as the granularity (i.e., synthesis interval) increases. This suggests that we can achieve a trade-off between the additional computational cost of parameter synthesis and the accuracy of expert coefficients (which directly impact task performance). Empirical results indicate that parameter synthesis (stages 1 & 2) incurs an average additional computation time of approximately 20 ms, while action generation (stage 3) requires about 30 ms. Results show that stage 1 & 2 can be executed at a much lower frequency than stage 3, at the cost of slightly decreased success rate.
>
> | Synthesize Interval | every 1 step | every 5 steps | every 10 steps | every 50 steps |
> | :----:| :----: | :----: |  :----: | :----: |
> | Sucess rate | $ 0.74 \pm 0.06 $ | $ 0.70 \pm 0.03 $ | $ 0.68\pm0.04$ |  $ 0.63 \pm 0.04$ |

---

> ### Author Response · Authors · 2025-11-30
> **Response (part 2)**
>
> We also investigate the applicability to ultra-long sequences. To this end, we concatenate the LIBERO-Object, Goal, and Spatial benchmark to form a task sequence with $K=30$. Following the empirical analysis in Section 5.2, we prune underactivated low-rank experts every 10 tasks to control the linear growth of the library. At the end of the 30-th task, the number of LoRA experts in each module (the elements of the six-dimensional vector in order representing the vision encoder, text encoder, state encoder, modality fusion module, temporal transformer, and action head) typically remains below 30, resulting in a compact library that exhibits effective knowledge sharing. DMPEL consistently outperforms FFT with experience replay in success rate, while incurring significantly lower computational and storage costs.
>
> | Method | Number of LoRA Experts and Parameters in Each Module |Trainable Parameters | Additional Storage for Replay |
> | :----:| :----:| :----: | :----: |
> DMPEL | [15,18,17,21,17,10], 10.7M (seed=100), [18,19,17,19,17,7], 11.1M (seed=200), [20,18,20,19,18,9], 11.7M (seed=300) | 1.2M | 0.06GB |
> ER |/ | 174.1M | 3.7GB |
>
> [1] Schmied, T., Hofmarcher, M., Paischer, F., Pascanu, R., & Hochreiter, S. (2023). Learning to modulate pre-trained models in rl. Advances in Neural Information Processing Systems, 36, 38231-38265.
>
> [2] Jung, D., Han, D., Bang, J., & Song, H. (2023). Generating instance-level prompts for rehearsal-free continual learning. In Proceedings of the IEEE/CVF International Conference on Computer Vision (pp. 11847-11857).
>
> [3] Wang, L., Xie, J., Zhang, X., Su, H., & Zhu, J. (2025). HIDE-PET: continual learning via hierarchical decomposition of parameter-efficient tuning. IEEE Transactions on Pattern Analysis and Machine Intelligence.

---

> ### Author Response · Authors · 2025-11-30
> **Response (part 3)**
>
> > Weakness 2: Limited task diversity in evaluation – experiments are restricted to LIBERO simulated environments, not real-world or cross-domain tasks.
>
> Ans: Thanks for your comment. We want to emphasize that LIBERO is a large-scale multi-modal benchmark that need complex behavior, hence our result is convincing. We further evaluate DMPEL with same hyperparameter on three tasks (Lift, Can, and Square) from Robomimic [1] , which formulate a task sequence with $K=3$. These tasks use same observation and action space, but with novel backgrounds and objects. The result is as follows:
>
> | Metircs | FWT | NBT | AUC |
> | :----:| :----:| :----: | :----: |
> | Robomimic | $ 0.54 \pm 0.05 $ | $ 0.10 \pm 0.02 $ | $ 0.54 \pm 0.02 $ |
>
> When the distribution shift becomes significantly larger, such as in sim2real settings, we anticipate that additional effort will be required for effective transfer. Specifically, we foresee the following challenges that need to be addressed for sim2real applications: (1) Differences in Visual Input. There are substantial discrepancies between images produced by simulation engines and those from real-world sensors, necessitating further fine-tuning of the visual encoder; (2) Differences Between Simulated and Real Robots. Even when using the same robot model (e.g., the Franka Emika Panda), minor differences can lead to varying end-effector behaviors, and hence we need demonstration collected on the real-robot; (3) Real-world stochsticity. The unpredictability of real-world environments adds complexity to the transfer process.
>
> One potential approach is to incorporate a post-fine-tuning step for the pretrained policy using real-world datasets before initiating lifelong adaptation in the real environment. Alternatively, we could start with more advanced VLA models (consisting of several billion parameters and pretrained on thousands of real-world trajectories with mixed embodiments) rather than the manuscript currently used, which is pretrained only on LIBERO-90. Given DMPEL's performance in the large-scale simulated benchmark LIBERO, we expect it to perform well during the lifelong adaptation phase. However, we would like to emphasize that the sim2real area have a wealth of related work and are generally orthogonal to lifelong learning approaches. In this manuscript, we focues on comparing different lifelong learning approaches with the same pretrained model and in the same benchmark. We plan to explore lifelong learning on real robots in future research.
>
> [1] Mandlekar, Ajay, et al. "What Matters in Learning from Offline Human Demonstrations for Robot Manipulation." Conference on Robot Learning. PMLR, 2022.

---

> ### Author Response · Authors · 2025-11-30
> **Response (part 4)**
>
> > Weakness 3: No comparisons with learning-based retrieval baselines like contrastive skill indexing or diffusion-based task retrievers.
>
> Ans: Thank you for your comment. Actually, in the manuscript we compare to the following baselines that involves learning-based low-level policy or LoRA expert retrieval: (1) LOTUS first extract temporal segment features by leveraging frozen DINOv2 and perform hierarchical clustering in the feature space. LOTUS trains a conditional Variational Auto-Encoder (cVAE) by minimizing an ELBO loss over demonstrations, in which the training supervision comes from the labels of skill indices from the clustering step and the subgoal embeddings obtained from encoding look-ahead images; (2) L2M maintains a learnable modulation pool with keys and associated modulators. The embedded history of state tokens serves as the query vector for query-key matching, which finally steers the pretrained policy's behavior. The training loss consists of two terms: the first is the behavior cloning loss, and the second is a surrogate loss that pulls the selected keys closer to their corresponding query features; (3) IsCiL establishes multifaceted prototypes through K-means clustering in the state space, where proper LoRA-based skills is retrieved upon current input state.
>
> Hence, we want to emphasize that these baselines involve encoding observations into a latent space and employing various learning methods to partition the latent space for skill indexing or expert retrieval through similarity search during inference. Nevertheless, we acknowledge that comparing our approach with the methods you mentioned could represent promising future directions: (1) contrastive learning for latent representation learning. We believe this could be a viable way to learn a skill similarity function, as demonstrated in [1]. However, [1] has only been evaluated in environments with low-dimensional state spaces and lacks open-source code, making replication on LIBERO within a short time challenging; (2) diffusion-based task retrievers. We understand this can be viewed as a form of generative replay, similar to methods discussed in [2] but applied in the robotics domain. Since DMPEL is an architectural lifelong learning method that utilizes LoRA modules to capture specific knowledge, we primarily compare it with methods such as TAIL, L2M, and IsCiL, which also leverage LoRA modules as their main technique. Results have already demonstrate the effectiveness of DMPEL over LoRA-based baselines. We additionally comparing against only one replay baseline, ER, in the main results, and we leave more comparison as our future work.
>
> [1] Choi, J., & Seo, S. W. (2025) Dynamic Contrastive Skill Learning with State-Transition Based Skill Clustering and Dynamic Length Adjustment. In The Thirteenth International Conference on Learning Representations.
>
> [2] Gao, R., & Liu, W. (2023). Ddgr: Continual learning with deep diffusion-based generative replay. In International Conference on Machine Learning (pp. 10744-10763). PMLR.

---

### Author Response · Authors · 2025-11-30
**General response**

Dear AC and reviewers,

We'd like to express our sincere gratitude for your time and effort. The insightful feedback prompted us to conduct additional experiments and improve the clarity. We believe these enhancement significantly strengthen our submission.

(1) Strengths recognized by the reviewers:

* The paper addresses an important problem (ijwr)
* The manuscript exhibits a complete and logical structure, and it is well written in a clear and coherent manner (vjgK)
* The framework allows fine-grained expert composition and knowledge sharing across tasks, achieves near-zero forgetting via expert coefficient replay, enabling stable and memory-efficient lifelong robot learning (1SdP, vjgK, hns8, ijwr)
* The paper provides extensive experiments that convincingly demontrate strong performance against baselines (hns8, ijwr)
* The paper offers extensive and clear ablation studies with analysis that demonstrate the functionality of each component (hns8, ijwr)
* The paper emphasizes reproducibility through a well-documented report with methodological details, hyperparameters, and supplementary material (ijwr)

(2) Questions raised during review, our efforts during rebuttal, and updates in the manuscript:

* We update our manuscipts with improved clarity, including clear definitions of all mathematical symbols and consistent formatting throughout the paper (vjgK)
* We update the t-sne figure in Section 5.2 with additional results of ER and EWC (hns8)
* We provide additional results on top-k mismatch between training and evaluation in Appendix B.1 (vjgK)
* We provide additional results on TAIL and IsCiL with rank increased by three times in Appendix B.1 (ijwr)
* We provide additional results on employing separate small models for each task in Appendix B.2 (vjgK)
* We provide visualization of the action space in Appendix B.3 (hns8)
* We update the expert activation visualization to provide more detailed analysis on knowledge reuse in Appendix B.3 (hns8)
* We provide additional results on the performance and the computational and storage cost analysis in the ultra-long task sequence in Appendix B.4 (1SdP, vjgK)
* We provide additional results on cross-domain transfer to Robomimic in Appendix B.5 (1SdP)
* We add discussions on the computational overhead (inference latency) (1SdP, hns8)
* We add related work (1SdP, vjgK)
* We add discussions on the relationship between multitask and lifelong learning (hns8, ijwr)
* We add a more thorough discussions on limitations and future directions in Appendix C (vjgK)

Best regards,

Authors

---

### Meta-Review · Area_Chair_yPDp · 2026-01-06

**Summary:**

DMPEL (Dynamic Mixture of Progressive Parameter-Efficient Expert Library) addresses lifelong robot learning under sequential tasks by growing a task-wise library of LoRA experts and learning a context-conditioned router that sparsely mixes the top-k experts at test time, thereby avoiding the unrealistic task-ID (known-task) assumption while enabling cross-task knowledge sharing. To mitigate catastrophic forgetting without expensive experience replay, it introduces expert coefficient replay, which stores and replays only low-dimensional pairs of (context embedding → routing coefficients) to regularize the router’s decisions on past tasks. Experiments on LIBERO lifelong manipulation suites (with ablations on top-k mixing, replay ratio, and inference cost trade-offs) report improved forward transfer and reduced forgetting relative to prior lifelong/PEFT baselines at a small trainable-parameter footprint, while discussing scalability and broader real-world generalization as future work.

**Reviewer Concerns:**

The reviewers agreed that the paper tackles an important problem, is clearly written and well organized, and presents strong experimental coverage. They also viewed the proposed dynamic expert composition as a compelling way to enable task-ID–free use of LoRA experts while still allowing parameter sharing across experts.

The reviewers expressed concerns regarding the paper’s positioning and conceptual novelty relative to prior PEFT and lifelong-learning work, the scalability and inference overhead of dynamic routing/mixing as the expert library grows, the breadth of evaluation beyond LIBERO, and the fairness/completeness of comparisons (including parameter-budget matching and retrieval-style baselines).

The authors responded by clarifying the relationship to existing methods, adding analyses and ablations on top-k routing, rank- and budget-matched baselines, and inference trade-offs (e.g., synthesis intervals and latency breakdowns), and by extending the empirical evidence with longer task sequences (including pruning) and additional cross-domain evaluation, while also improving notation and expanding the limitations/future-work discussion.

**Reviewer Scores:**

Three of the reviewers scored the paper as marginally below the acceptance threshold, while one scored it as marginally above the threshold. I think the authors' responses address some of the reviewers' concerns, but the paper remains borderline.

However, in light of recent related work, most notably:

Ge et al., Dynamic Mixture of Curriculum LoRA Experts for Continual Multimodal Instruction Tuning (ICML 2025),

the paper’s novelty appears limited, with the main differences largely confined to implementation choices and the application domain, which ultimately weakens the submission’s contribution.

For these reasons, while I acknowledge the paper’s contributions, I lean toward rejecting the submission.

---

### Decision · Program_Chairs · 2026-01-26

Reject